# DOUBLE MOMENTUM METHOD FOR LOWER-LEVEL CONSTRAINED BILEVEL OPTIMIZATION

## ABSTRACT

Bilevel optimization (BO) has recently gained prominence in many machine learning applications due to its ability to capture the nested structure inherent in these problems. Recently, many gradient-based methods have been proposed as effective solutions for solving large-scale problems. However, current methods for the lower-level constrained bilevel optimization (LCBO) problems lack a solid analysis of the convergence rate. What's worse, existing methods require either double-loop updates, which are sometimes less efficient. To solve this problem, in this paper, we propose a novel *single-loop single-timescale* method with theoretical guarantees for LCBO problems. Specifically, we leverage the Gaussian smoothing to design an approximation of the hypergradient. Then, using this hypergradient, we propose a *single-loop single-timescale* algorithm based on the double-momentum method and adaptive step size method. Theoretically, we demonstrate that our methods can return a stationary point with $\tilde{\mathcal{O}}(d_2^2 \epsilon^{-4})$ iterations. In addition, experiments on two applications also demonstrate the superiority of our proposed method.

## 1 INTRODUCTION

Bilevel optimization (BO) (Bard, 2013; Colson et al., 2007) plays a central role in various significant machine learning applications including hyper-parameter optimization (Pedregosa, 2016; Bergstra et al., 2011; Bertsekas, 1976), meta-learning (Feurer et al., 2015; Franceschi et al., 2018; Rajeswaran et al., 2019), reinforcement learning (Hong et al., 2020; Konda & Tsitsiklis, 2000). Generally speaking, the BO can be formulated as follows,

$$\min_{x \in \mathcal{X}} F(x) = f(x, y^*(x)) \quad s.t. \quad y^*(x) = \arg\min_{y \in \mathcal{Y}} g(x, y), \tag{1}$$

where $\mathcal{X}$ and $\mathcal{Y}$ are convex subsets in $\mathbb{R}^{d_1}$ and $\mathbb{R}^{d_2}$, respectively. It involves a competition between two parties or two objectives, and if one party makes its choice first, it will affect the optimal choice of the other party.

Recently, gradient-based methods have shown great effectiveness in solving various large-scale bilevel optimization problems, where there is no constraint in the lower-level objective, *i.e.*, $\mathcal{Y} = \mathbb{R}^{d_2}$. Specifically, Franceschi et al. (2017); Pedregosa (2016); Ji et al. (2021) proposed several double-loop algorithms to solve the BO problems. They first apply the gradient methods to approximate the solution to the lower-level problem and then implicit differentiable methods (Pedregosa, 2016; Ji et al., 2021) or explicit differentiable methods (Franceschi et al., 2017) can be used to approximate the gradient of the upper-level objective w.r.t $x$, namely hypergradient, to update $x$. However, in some real-world applications, such as in a sequential game, the problems must be updated at the same time Hong et al. (2020), which makes these methods unsuitable. To solve this problem, Hong et al. (2020) propose a single-loop two-timescale method, which updates $y$ and $x$ alternately with stepsize $\eta_y$ and $\eta_x$, respectively, designed with different timescales as $\lim_{k \to \infty} \eta_x/\eta_y = 0$. However, due to the nature of two-timescale updates, it incurs the sub-optimal complexity $\mathcal{O}(\epsilon^{-5})$ (Chen et al., 2021). To further improve the efficiency, Huang & Huang (2021); Khanduri et al. (2021); Chen et al. (2021); Guo et al. (2021a) proposed single-loop single-timescale methods, where $\eta_x/\eta_y$ is a constant. These methods have the complexity of $\tilde{\mathcal{O}}(\epsilon^{-4})$ ($\tilde{\mathcal{O}}$ means omitting logarithmic factors) or better ($\tilde{\mathcal{O}}(\epsilon^{-3})$) to achieve the stationary point. However, all these methods are limited to the bilevel optimization problem with unconstrained lower-level problems and require the upper-level objective function to

Table 1: Several representative hypergradient approximation methods for the lower-level constrained BO problem. The final column shows iteration numbers to find a stationary point $\|\nabla F(x)\| \leq \epsilon$ (Here $d_2$ denotes the dimension of the lower-level variable, $\delta$ is the smoothing parameter.).

| Method | Reference | Loop | Timescale | LL Constraint | UL Constraint | Iterations (Upper) |
|--------|-----------|------|-----------|---------------|---------------|--------------------|
| AiPOD | (Xiao et al., 2023) | Double | $\times$ | Affine sets | Affine sets | $\tilde{\mathcal{O}}(\epsilon^{-4})$ |
| IG-AL | (Tsaknakis et al., 2022) | Double | $\times$ | Half space | $\times$ | $\times$ |
| IAPTT-GM | (Liu et al., 2021) | Double | $\times$ | Convex set | Convex set | $\times$ |
| RMD-PCD | (Bertrand et al., 2022) | Double | $\times$ | Norm set | $\times$ | $\times$ |
| JaxOpt | (Blondel et al., 2022) | Double | $\times$ | Convex set | $\times$ | $\times$ |
| DMLCBO | Ours | Single | Single | Convex set | Convex set | $\tilde{\mathcal{O}}(d_2^2\epsilon^{-4})$ |

be differentiable. They cannot be directly applied when constraints are present in the lower-level optimization, i.e., $\mathcal{Y} \neq \mathbb{R}^{d_2}$, as the upper-level objective function is naturally non-smooth (Xu & Zhu, 2023).

To solve the lower-level constrained bilevel optimization problem, various methods have been proposed to approximate the hypergradient, as shown in Table 1. Specifically, Xiao et al. (2023) reformulate the affine-constrained lower-level problem into an unconstrained problem, and then solve the new lower-level problem and use the implicit differentiable method to approximate the hyper-gradient. However, their convergence analysis only focuses on the affine-constrained problem and cannot be extended to a more general case. Tsaknakis et al. (2022) solve the inner problem with projection gradient and use the implicit differentiable method to approximate the hyper-gradient for the half-space-constrained BO problem. However, they only give the asymptotic convergence analysis for this special case. Since many methods of calculating the Jacobian of the projection operators have been proposed (Martins & Astudillo, 2016; Djolonga & Krause, 2017; Blondel et al., 2020; Niculae & Blondel, 2017; Vaiter et al., 2013; Cherkaoui et al., 2020), the explicit or implicit methods can also be used to approximate the hypergradient in the LCBO problems, such as Liu et al. (2021); Bertrand et al. (2020; 2022); Blondel et al. (2022). However, these methods lack solid convergence analysis. What's worse, these methods can not be utilized to solve the sequential game which is mentioned above. Therefore, it is still an open challenge to design a *single-loop single-timescale* method with convergence rate analysis for the lower-level constrained bilevel optimization problems.

To overcome these problems, we propose a novel *single-loop single-timescale* method with a convergence guarantee for the lower-level constrained BO problems. Specifically, we leverage the Gaussian smoothing to design a hypergradient. Then, using our new hypergradient, we propose a *single-loop single-timescale* algorithm based on the double-momentum method and adaptive step size method to update the lower- and upper-level variables simultaneously. Theoretically, we prove our methods can return a stationary point with $\tilde{\mathcal{O}}(d_2^2\epsilon^{-4})$ iterations. In addition, we compare our method with three state-of-the-art methods for the lower-level constrained BO problems in data hypercleaning and training data poison attack. The experimental results in these two applications demonstrate the efficiency and effectiveness of our proposed method.

We summarized our contributions as follows.

1. We propose a new method to approximate the hypergradient using the Gaussian smoothing for the constrained bilevel problem. Using this hypergradient, we propose a *single-loop single-timescale* algorithm for the lower-level constrained BO problems, while existing methods are all double-loop methods.

2. Existing methods for solving the lower-level constrained BO problems usually lack theoretical analysis on convergence rate. We prove our methods can return a stationary point with $\tilde{\mathcal{O}}(d_2^2\epsilon^{-4})$ iterations.

3. We compare our method with several state-of-the-art methods for the lower-level constrained BO problems on two applications. The experimental results demonstrate the superiority of our proposed method in terms of training time and accuracy.

## 2 PRELIMINARIES

**Notations.** Here we give several important notations used in this paper. $\| \cdot \|$ denotes the $\ell_2$ norm for vectors and spectral norm for matrices. $I_d$ denotes a $d$-dimensional identity matrix. $A^\top$ denotes transpose of matrix $A$. Given a convex set $\mathcal{X}$, we define a projection operation to $\mathcal{X}$ as $\mathcal{P}_\mathcal{X}(x') = \arg\min_{x \in \mathcal{X}} 1/2\|x - x'\|^2$.

### 2.1 PROBLEM SETTING OF THE LOWER-LEVEL CONSTRAINED BILEVEL OPTIMIZATION PROBLEM

In this paper, we consider the following BO problems where both upper- and lower-level problems have the convex constraints,

$$\min_{x \in \mathcal{X} \subseteq \mathbb{R}^{d_1}} F(x) = f(x, y^*(x)) \quad s.t. \quad y^*(x) = \arg\min_{y \in \mathcal{Y} \subseteq \mathbb{R}^{d_2}} g(x, y). \tag{2}$$

Then, we introduce several mild assumptions on the Problem (2).

**Assumption 1.** *The upper-level function $f(x, y)$ satisfies the following conditions:*

1. *$\nabla_x f(x, y)$ is $L_f$-Lipschitz continuous w.r.t. $(x, y) \in \mathcal{X} \times \mathcal{Y}$ and $\nabla_y f(x, y)$ is $L_f$-Lipschitz continuous w.r.t $(x, y) \in \mathcal{X} \times \mathcal{Y}$, where $L_f \geq 0$.*

2. *For any $x \in \mathcal{X}$ and $y \in \mathcal{Y}$, we have $\|\nabla_y f(x, y)\| \leq C_{fy}$.*

**Assumption 2.** *The lower-level function $g(x, y)$ satisfies the following conditions:*

1. *For any $x \in \mathcal{X}$ and $y \in \mathcal{Y}$, $g(x, y)$ is twice continuously differentiable in $(x, y)$.*

2. *Fix $x \in \mathcal{X}$, $\nabla_y g(x, y)$ is $L_g$-Lipschitz continuous w.r.t $y \in \mathcal{Y}$ for some $L_g \geq 1$.*

3. *Fix $x \in \mathcal{X}$, for any $y \in \mathcal{Y}$, $g(x, y)$ is $\mu_g$-strongly-convex in $y$ for some $\mu_g > 0$.*

4. *$\nabla^2_{xy} g(x, y)$ is $L_{gxy}$-Lipschitz continuous w.r.t $(x, y) \in \mathcal{X} \times \mathcal{Y}$ and $\nabla^2_{yy} g(x, y)$ is $L_{gyy}$-Lipschitz continuous w.r.t $(x, y) \in \mathcal{X} \times \mathcal{Y}$, where $L_{gxy} \geq 0$ and $L_{gyy} \geq 0$.*

5. *For any $x \in \mathcal{X}$ and $y \in \mathcal{Y}$, we have $\|\nabla^2_{xy} g(x, y)\| \leq C_{gxy}$.*

These assumptions are commonly used in bilevel optimization problems (Ghadimi & Wang, 2018; Hong et al., 2020; Ji et al., 2021; Chen et al., 2021; Khanduri et al., 2021; Guo et al., 2021a).

### 2.2 REVIEW OF UNCONSTRAINED BILEVEL OPTIMIZATION METHODS

For the upper-level objective, we can naturally derive the following gradient w.r.t $x$ using the chain rule (which is defined as hypergradient),

$$\nabla F(x) = \nabla_x f(x, y^*(x)) + (\nabla y^*(x))^\top \nabla_y f(x, y^*(x)). \tag{3}$$

Obviously, the crucial problem of obtaining the hypergradient is calculating $\nabla y^*(x)$. If the lower-level problem is unconstrained, using the implicit differentiation method and the optimal condition $\nabla_y g(x, y^*(x)) = 0$, it is easy to show that for a given $x \in \mathbb{R}^{d_1}$, the following equation holds (Ghadimi & Wang, 2018; Hong et al., 2020; Ji et al., 2021; Chen et al., 2021; Khanduri et al., 2021)

$$\nabla y^*(x) = -[\nabla^2_{yy} g(x, y^*(x))]^{-1} \nabla^2_{yx} g(x, y^*(x)). \tag{4}$$

Substituting $\nabla y^*(x)$ into $\nabla F(x)$, we can obtain the hypergradient. Then, we update $x$ and $y$ alternately using the gradient method.

### 2.3 HYPERGRDIENT OF LOWER-LEVEL CONSTRAINED BILEVEL OPTIMIZATION PROBLEM
For the constrained lower-level problem, one common method is to use the projection gradient method, which has the following optimal condition,

$$y^*(x) = \mathcal{P}_\mathcal{Y}(y^*(x) - \eta \nabla_y g(x, y^*(x))), \tag{5}$$

where $\eta > 0$ denotes the step-size. By Rademacher's theorem (Federer, 1969), the projection operator is differentiable almost everywhere. Recently, based on Assumption 3, Blondel et al. (2022); Bertrand et al. (2020; 2022) derive the following hypergradient for the lower-level constrained BO problem,

$$
\begin{aligned}
\nabla F(x) = {}& \nabla_x f(x, y^*(x)) \\
& - \eta \nabla^2_{xy} g(x, y^*(x)) H^\top \left[ I_{d_2} - (I_{d_2} - \eta \nabla^2_{yy} g(x, y^*(x))) \cdot H^\top \right]^{-1} \nabla_y f(x, y^*(x)), \quad (6)
\end{aligned}
$$

where $H = \nabla \mathcal{P}_\mathcal{Y}(z^*)$ and $z^* = y^*(x) - \eta \nabla_y g(x, y^*(x))$. Note that the gradient of the projection operator can be easily obtained using the method in (Martins & Astudillo, 2016; Djolonga & Krause, 2017; Blondel et al., 2020; Niculae & Blondel, 2017; Vaiter et al., 2013; Cherkaoui et al., 2020).

**Assumption 3.** *a) If projection operator has a closed-form solution and $z^* = y^*(x) - \eta \nabla_y g(x, y^*(x))$ is not on the boundary of the constraint, then $\mathcal{P}_\mathcal{Y}(z^*)$ is continuously differentiable in a neighborhood of $z^*$. In addition, in the neighborhood of $z^*$, $\mathcal{P}_\mathcal{Y}(z^*)$ has Lipshitz continuous gradienet with constant L. b) $y^*(x)$ is continuously differentiable on a neighborhood of $x$.*

In many complicated machine learning problems, the probability that $z^*$ falls exactly on the constraint boundary is very low. This means that, in many problems, when we obtain the optimal solution, the constraint may be not active or $\|\nabla_y g(x, y)\| > 0$. In these two cases, the projection operator can be viewed as differentiable in the neighborhood of $z^*$.

To derive a convergence rate of finding the stationary point, traditional methods (Khanduri et al., 2021; Hong et al., 2020; Chen et al., 2021; Huang & Huang, 2021) need the hypergradient $\nabla F(x)$ to be Lipschitz continuous (*i.e.*, $\|\nabla F(x_1) - \nabla F(x_2)\| \leq L_F \|x_1 - x_2\|$, where $L_F \geq 0$). This condition can be easily obtained if the lower-level problem is unconstrained. However, if the lower-level objective is constrained, obtaining the above condition needs the projection operator to be smooth (*i.e.*, $\|\nabla \mathcal{P}_\mathcal{Y}(z_1^*) - \nabla \mathcal{P}_\mathcal{Y}(z_2^*)\| \leq L_P \|z_1^* - z_2^*\|$, where $L_P \geq 0$, $z_1^* = y(x_1) - \eta \nabla g(x_1, y^*(x_1))$ and $z_2^* = y(x_2) - \eta \nabla g(x_2, y^*(x_2)))$. Obviously, most projection operators are nonsmooth (Martins & Astudillo, 2016; Djolonga & Krause, 2017; Blondel et al., 2020; Niculae & Blondel, 2017; Vaiter et al., 2013; Cherkaoui et al., 2020) which makes the above condition not satisfied and difficult to obtain the convergence analysis.

## 3 PROPOSED METHOD

In this section, we propose a new method to approximate the hypergradient using Gaussian smoothing that makes convergence analysis possible. Then, equipped with this hypergradient, we propose our *single-loop single-timescale* method to find a stationary point of the lower-level constrained bilevel problem.

### 3.1 GAUSSIAN SMOOTHING

Inspired by the strong ability of Gaussian smoothing to deal with nonsmooth problems, in this subsection, we use this method to handle the non-smoothness of the projection operator. Given a non-expansive projection operator (Moreau, 1965) $\mathcal{P}_\mathcal{Y}(z)$ and a distribution $\mathbb{P} = \mathcal{N}(0, I_{d_2})$, we define the smoothing function as $\mathcal{P}_{\mathcal{Y}\delta}(z) = \mathbb{E}_{u \sim \mathbb{P}}[\mathcal{P}_\mathcal{Y}(z + \delta u)]$. Then, we have the following proposition.

**Proposition 1.** *Let $\mathcal{P}_{\mathcal{Y}\delta}(z) = \mathbb{E}_{u \sim \mathbb{P}}[\mathcal{P}_\mathcal{Y}(z + \delta u)]$ where $\mathbb{P} = \mathcal{N}(0, I_{d_2})$. Since that $\mathcal{P}_\mathcal{Y}$ is non-expansive, we have (1) $\|\nabla \mathcal{P}_{\mathcal{Y}\delta}(z^*) - \nabla \mathcal{P}_\mathcal{Y}(z^*)\| \leq \dfrac{\delta L}{2}(d_2 + 3)^{3/2}$ on the neighborhood of $z^*$, where $z^* = y^*(x) - \eta \nabla_y g(x, y^*(x))$, and (2) $\mathcal{P}_{\mathcal{Y}\delta}(z)$ is differentiable and $1$-Lipschitz continuous with the $L_{P_\delta} = 2\sqrt{d}/\delta$-Lipschitz gradient.*

Using this Gaussian smoothing function, we can approximate the hypergradient as follows,

$$
\begin{aligned}
& \nabla F_\delta(x) \\
= {}& -\eta \nabla^2_{xy} g(x, y^*(x)) \nabla \mathcal{P}_{\mathcal{Y}\delta}(z^*)^\top \left[ I_{d_2} - (I_{d_2} - \eta \nabla^2_{yy} g(x, y^*(x))) \nabla \mathcal{P}_{\mathcal{Y}\delta}(z^*)^\top \right]^{-1} \nabla_y f(x, y^*(x)) \\
& + \nabla_x f(x, y^*(x)). \quad (7)
\end{aligned}
$$

where $z^* = y^*(x) - \eta \nabla_y g(x, y^*(x))$. For this hypergradient estimation, we have the following conclusion.

**Lemma 1.** *Under Assumptions 1, 2, we have $\nabla F_\delta(x)$ is Lipschitz continuous w.r.t $x$.*

**Remark 1.** *Lemma 1 indicates we can discuss the convergence performance on the approximated function $F_\delta(x)$ whose gradient is $\nabla F_\delta(x)$. Once we can obtain the errors between $\nabla F_\delta(x)$ and $\nabla F(x)$, we can obtain the convergence result of the original problem.*

### 3.2 Approximation of hypergradient

To calculate the hypergradient $\nabla F_\delta(x)$, we need the optimal solution $y^*(x)$, which is usually difficult. What's worse, we need to compute the matrix inverse, which has $\mathcal{O}(d_2^3)$ time complexity. Thus, in this subsection, we discuss how to calculate the hypergradient approximation in an efficient method.

As shown in (Huang & Huang, 2021; Khanduri et al., 2021; Chen et al., 2021; Hong et al., 2020), one proper method is to use $y$ to approximate $y^*(x)$. Using this method, we can approximate the hypergradient at $(x, y)$ as follows,

$$\nabla f_\delta(x, y) = \nabla_x f(x, y) - \eta \nabla_{xy}^2 g(x, y) \nabla \mathcal{P}_{\mathcal{Y}\delta}(z)^\top \left[ I_{d_2} - (I_{d_2} - \eta \nabla_{yy}^2 g(x, y)) \nabla \mathcal{P}_{\mathcal{Y}\delta}(z)^\top \right]^{-1} \nabla_y f(x, y).$$

where $z = y - \eta \nabla_y g(x, y)$

To efficiently approximate the matrix inverse, the following well-known result can be used in our method (Ghadimi & Wang, 2018; Meyer, 2000).

**Lemma 2.** *Let $A \in \mathbb{R}^{d \times d}$ be a matrix with $\|A\| < 1$, then we have $(I_d - A)^{-1} = \sum_{i=0}^{\infty} A^i$.*

To utilize this lemma to approximate $\left[ I_{d_2} - (I_{d_2} - \eta \nabla_{yy}^2 g(x, y)) \nabla \mathcal{P}_{\mathcal{Y}\delta}(z)^\top \right]^{-1}$, one crucial step is to ensure $\|(I_{d_2} - \eta \nabla_{yy}^2 g(x, y)) \nabla \mathcal{P}_{\mathcal{Y}\delta}(z)^\top\| \le 1$. Fortunately, since the projection operator is non-expansive, we have $\|\mathcal{P}_{\mathcal{Y}\delta}(z)\| \le 1$ (See Proposition 1). Then, setting $\eta \le \dfrac{1}{L_g}$, we can easily obtain $\|I_{d_2} - \eta \nabla_{yy}^2 g(x, y)\| \le 1$. Thus, we can further approximate the hypergradient as follows,

$$\bar{\nabla} f_\delta(x, y) = \nabla_x f(x, y) - \eta \nabla_{xy}^2 g(x, y) \nabla \mathcal{P}_{\mathcal{Y}\delta}(z)^\top \sum_{i=0}^{Q-1} \left( (I_{d_2} - \eta \nabla_{yy}^2 g(x, y)) \nabla \mathcal{P}_{\mathcal{Y}\delta}(z)^\top \right)^i \nabla_y f(x, y).$$

In addition, we can use the following unbiased estimator of the gradient $\nabla \mathcal{P}_{\mathcal{Y}\delta}(z)$ to replace $\nabla \mathcal{P}_{\mathcal{Y}\delta}(z)$,

$$\bar{H}(z; u) = \frac{1}{\delta} \left( \mathcal{P}_{\mathcal{Y}}(z + \delta u) - \mathcal{P}_{\mathcal{Y}}(z) \right) u^\top. \tag{8}$$

For $\bar{H}(z; u)$, we have the following conclusion.

**Lemma 3.** *We have $\mathbb{E}_u \left[ \bar{H}(z; u) \right] = \nabla \mathcal{P}_{\mathcal{Y}\delta}(z)$ and $\mathbb{E}_u \left[ \left\| \bar{H}(z; u) - \nabla \mathcal{P}_{\mathcal{Y}\delta}(z) \right\|^2 \right] \le (d_2 + 4)^2 + 2$.*

To further reduce the complexity caused by calculating multiple Jacobian-vector products, we can introduce an additional stochastic layer on the finite sum. Specifically, assume we have a parameter $Q > 0$ and a collection of $Q + 2$ independent samples $\bar{\xi} := \{u^0, \cdots, u^{Q-1}, c(Q)\}$, where $c(Q) \sim \mathcal{U} \{0, \cdots, Q - 1\}$. Then, we can approximate the gradient as follows,

$$\bar{\nabla} f_\delta(x, y; \bar{\xi})$$
$$= \nabla_x f(x, y) - \eta Q \nabla_{xy}^2 g(x, y) \bar{H}(z; u^0)^\top \prod_{i=1}^{c(Q)} \left( (I_{d_2} - \eta \nabla_{yy}^2 g(x, y)) \bar{H}(z; u^i)^\top \right) \nabla_y f(x, y), \tag{9}$$

where we have used the the convention $\prod_{i=1}^{0} A = I$. We can conclude that the bias of the gradient estimator $\bar{\nabla} f_\delta(x, y; \bar{\xi})$ decays exponentially fast with $Q$, as summarized below:

**Lemma 4.** *Under Assumptions 1, 2and Lemma 3, setting $\dfrac{1}{\mu_g} (1 - \dfrac{1}{d_2 + 4}) \le \eta \le \dfrac{1}{\mu_g}$, for any $x \in \mathbb{R}^{d_1}$, $y \in \mathcal{Y}$, we have $\left\| \nabla f_\delta(x, y) - \mathbb{E}[\bar{\nabla} f_\delta(x, y; \bar{\xi})] \right\| \le \dfrac{C_{gxy} C_{fy}}{\mu_g} (1 - \eta \mu_g)^Q$. Furthermore, the variance of $\bar{\nabla} f_\delta(x, y; \bar{\xi})$ is bounded as $\mathbb{E} \left[ \left\| \bar{\nabla} f_\delta(x, y; \bar{\xi}) - \mathbb{E} \left[ \bar{\nabla} f_\delta(x, y; \bar{\xi}) \right] \right\|^2 \right] \le \sigma_f^2(d_2)$, where $\sigma_f^2(d_2)$ is defined in Appendix D.*

---

**Algorithm 1** DMLCBO

---

**Input:** Initialize $x_1 \in \mathcal{X}$, $y_1 \in \mathcal{Y}$, $w_1 = \bar{\nabla} f_\delta(x_1, y_1; \bar{\xi}_1)$, $v_1 = g(x_1, y_1)$, $\eta_k$, $\tau$, $\gamma$, $\beta$, $\alpha$, $Q$ and $\eta$.
 1: **for** $k = 1, \cdots, K$ **do**
 2:     Update $x_{k+1} = (1 - \eta_k)x_k + \eta_k \mathcal{P}_\mathcal{X}(x_k - \frac{\gamma}{\sqrt{\|w_k\|} + G_0} w_k)$.
 3:     Update $y_{k+1} = (1 - \eta_k)y_k + \eta_k \mathcal{P}_\mathcal{Y}(y_k - \frac{\tau}{\sqrt{\|v_k\|} + G_0} v_k)$
 4:     Calculate the hyper-gradient $\bar{\nabla} f_\delta(x_{k+1}, y_{k+1}; \bar{\xi}_{k+1})$ according to Eqn. (9).
 5:     Update $w_{k+1} = (1 - \alpha)w_k + \alpha \bar{\nabla} f_\delta(x_{k+1}, y_{k+1}; \bar{\xi}_{k+1})$.
 6:     Update $v_{k+1} = (1 - \beta)v_k + \beta \nabla_y g(x_{k+1}, y_{k+1})$.
 7: **end for**
**Output:** $x_r$ where $r \in \{1, \cdots, K\}$ is uniformly sampled.

---

### 3.3 DOUBLE-MOMENTUM METHOD FOR LOWER-LEVEL CONSTRAINED BILEVEL OPTIMIZATION

Equipped with the hypergradient $\bar{\nabla} f_\delta(x, y; \bar{\xi})$, our next endeavor is to design a *single-loop single-timescale* algorithm to solve the constrained bilevel optimization problem (2). Our main idea is to adopt the double-momentum-based method and adaptive step-size method developed in (Huang & Huang, 2021; Khanduri et al., 2021; Shi et al., 2022). Our algorithm is summarized in Algorithm 1. Since we use the double-momentum method to solve the lower-level constrained bilevel optimization problem, we denote our method as DMLCBO.

Define $\alpha \in (0, 1)$ and $\beta \in (0, 1)$. For the lower-level problem, we can utilize the following projected gradient method with the momentum-based gradient estimator and adaptive step size to update $y$,

$$y_{k+1} = (1 - \eta_k)y_k + \eta_k \mathcal{P}_\mathcal{Y}(y_k - \frac{\tau}{\sqrt{\|v_k\|} + G_0} v_k), \quad v_{k+1} = (1 - \beta)v_k + \beta \nabla_y g(x_k, y_k),$$

where $\eta_k > 0$, $\tau > 0$. $G_0 > 0$ is used to avoid to prevent the molecule from being equal to 0. Here, we initialize $v_1 = g(x_1, y_1)$. Similarly, for the upper-level problem, we can utilize the following gradient method with the momentum-based gradient estimator and adaptive step size to update $x$,

$$x_{k+1} = (1 - \eta_k)x_k + \eta_k \mathcal{P}_\mathcal{X}(x_k - \frac{\gamma}{\sqrt{\|w_k\|} + G_0} w_k), \quad w_{k+1} = (1 - \alpha)w_k + \alpha \bar{\nabla} f_\delta(x_k, y_k; \bar{\xi}_k),$$

and we initialize $w_1 = \bar{\nabla} f_\delta(x_1, y_1; \bar{\xi}_1)$.

## 4 CONVERGENCE ANALYSIS

In this section, we discuss the convergence performance of our DMLCBO (All the detailed proofs are presented in our Appendix). We follow the theoretical analysis framework in (Huang & Huang, 2021; Huang et al., 2020; Shi et al., 2022; Khanduri et al., 2021) (For easy understanding, we also provide a route map of the analysis in Appendix E). Before proceeding to the main results, we present an additional assumption and a useful lemma.

**Assumption 4.** *We have $c_l \leq 1/(\sqrt{\|v_k\|} + G_0) \leq c_u$ and $c_l \leq 1/(\sqrt{\|w_k\|} + G_0) \leq c_u$.*

The above assumption is used to bind the adaptive terms in our method which has been widely used in convergence analysis (Shi et al., 2022; Guo et al., 2021b). Since the gradient estimation is bounded, we can easily bound $v$ and $u$ for fixed momentum parameters. Even if the condition in Assumption 4 is not satisfied, we can also use the clipping method to make $v$ and $u$ bounded and finally make Assumption 4 hold.

Then, we discuss the metric used to evaluate the convergence performance. When dealing with constrained optimization problems, one proper method is to utilize $\mathcal{G}(x_k, \nabla F(x_k), \hat{\gamma}) = \frac{1}{\hat{\gamma}}(x_k - \mathcal{P}_\mathcal{X}(x_k - \hat{\gamma}\nabla F(x_k)))$ as a metric to assess the convergence performance, where $\hat{\gamma} = \gamma/(\sqrt{\|w_k\|} + G_0)$. We can bound $\mathcal{G}(x_k, \nabla F(x_k), \hat{\gamma})$ in following lemma,

**Lemma 5.** *Define metric $\mathcal{G}(x_k, \nabla F(x_k), \hat{\gamma}) = \frac{1}{\hat{\gamma}}(x_k - \mathcal{P}_\mathcal{X}(x_k - \hat{\gamma}\nabla F(x_k)))$, and let $\hat{\gamma} = \frac{\gamma}{\sqrt{\|w_k\|} + G_0}$ and $L_0 = \max(L_1(\frac{\sqrt{d_2}}{\delta}), L_2(\frac{\sqrt{d_2}}{\delta}))$ and $R_k = \nabla f_\delta(x, y) - \mathbb{E}[\bar{\nabla} f_\delta(x, y; \bar{\xi})]$, we*

*have*

$$\|\mathcal{G}\left(x_k, \nabla F(x_k), \hat{\gamma}\right)\| \leq \mathcal{M}_k, \tag{10}$$

*where* $\mathcal{M}_k = \|w_k - \nabla f(x_k, y_k) - R_k\| + \|R_k\| + L_0\|y^*(x_k) - y_k\| + \dfrac{1}{\gamma c_l}\|x_k - \tilde{x}_{k+1}\| + \dfrac{\delta L}{2\mu_g}C_{gxy}(d_2 + 3)^{3/2}C_{fy}(1 + \dfrac{1}{\mu_g}(1 - \eta\mu_g))$ *and* $\tilde{x}_{k+1} = \mathcal{P}_{\mathcal{X}}(x_k - \dfrac{\gamma}{\sqrt{\|w_k\|} + G_0}w_k).$

Hence, using the above lemma, we can define a new metric $\mathcal{M}_k$ to discuss the convergence of our method. In the scenario where $\mathcal{M}_k$ tends towards zero, we can easily observe that $\|\mathcal{G}(x_k, \nabla F(x_k), \hat{\gamma})\|$ also tends towards zero.

Then, turning back to the convergence analysis, using the above assumptions and lemmas, we can obtain the following theorem (For ease of reading, some parameter Settings are omitted here. The specific parameters can be found in Appendix G.2):

**Theorem 1.** *Under Assumptions 1, 2 4 and Lemma 3, setting* $Q = \dfrac{1}{\mu_g\eta}\ln\dfrac{C_{gxy}C_{fy}K}{\mu_g}$, $\eta_k = \dfrac{t}{(m+k)^{1/2}}$, $\alpha = c_1\eta_k$, $\beta = c_2\eta_k$, $t > 0$, *we have*

$$\frac{1}{K}\sum_{k=1}^{K}\mathbb{E}[\frac{1}{2}\mathcal{M}_k] \leq \frac{2m^{1/4}\sqrt{G}}{\sqrt{K}t} + \frac{2\sqrt{G}}{(Kt)^{1/4}} + \frac{\delta L}{4\mu_g}C_{gxy}(d_2 + 3)^{3/2}C_{fy}(1 + \frac{1}{\mu_g}(1 - \eta\mu_g)), \tag{11}$$

*where* $c_1, c_2$ *and* $G$ *are defined in the appendix.*

**Remark 2.** *Let* $t = \mathcal{O}(1)$, $a = 0$, $m = \mathcal{O}(1)$ *and* $\ln(m + K) = \tilde{\mathcal{O}}(1)$, *we have* $\sqrt{G} = \tilde{\mathcal{O}}(\sqrt{d_2})$. *In addition, let* $\delta = \mathcal{O}(\epsilon d_2^{-3/2})$. *Thus, our proposed DMLCBO can converge to a stationary point at the rate of* $\tilde{\mathcal{O}}(\dfrac{\sqrt{d_2}}{K^{1/4}})$. *Then, let* $a = \dfrac{1}{8}$, $r$ *randomly sampled from* $\{0, 1, \cdots, K\}$, *and* $\mathbb{E}[\dfrac{1}{2}\mathcal{M}_r] = \dfrac{1}{K}\sum_{k=1}^{K}\mathbb{E}[\dfrac{1}{2}\mathcal{M}_k] = \tilde{\mathcal{O}}(\dfrac{\sqrt{d_2}}{K^{1/4}}) \leq \epsilon$, *we have* $K = \tilde{\mathcal{O}}(\dfrac{d_2^2}{\epsilon^4})$.

## 5 EXPERIMENTS

In this section, we compare the performance of our method with SOTA methods for LCBO in two applications. (Detailed settings are given in Appendix.)

### 5.1 BASELINES

In this paper, we compare our method with the following state-of-the-art LCBO methods.

1. **PDBO**. The method proposed in (Sow et al., 2022) which uses the value function method to solve the bilevel optimization problem.
2. **RMD-PCD**. The method proposed in (Bertrand et al., 2022) which uses the reverse method to calculate the hyper-gradient.
3. **Approx**. The method proposed in (Pedregosa, 2016) which solves a linear optimization problem to calculate the hypergradient.

For the fairness of the experiments, we use the same rules to update $x$, and we implement all the methods by Pytorch (Paszke et al., 2019). Since JaxOpt (Blondel et al., 2022) is implemented by JAX (Bradbury et al., 2018), for a fair comparison, we use Approx with the Jacobian calculating methods in (Martins & Astudillo, 2016; Djolonga & Krause, 2017; Blondel et al., 2020; Niculae & Blondel, 2017; Vaiter et al., 2013; Cherkaoui et al., 2020) as a replacement of JaxOpt, which uses the same method to calculate the hypergradient. We run all the methods 10 times on a PC with four 1080Ti GPUs.

### 5.2 APPLICATIONS

**Data hyper-cleaning.** In this experiment, we evaluate the performance of all the methods in the application named data hyper-cleaning. In many real-world applications, the training set and testing

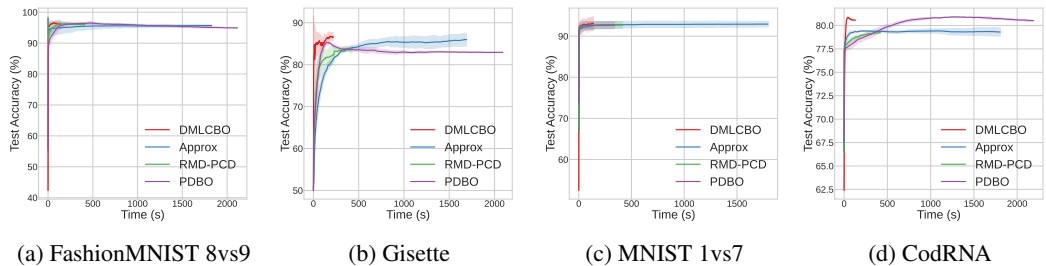

| (a) FashionMNIST 8vs9 | (b) Gisette | (c) MNIST 1vs7 | (d) CodRNA |

Figure 1: Test accuracy against training time of all the methods in data hyper-cleaning.

Table 2: Test accuracy (%) with standard variance of all the methods in data hyper-cleaning. The second column shows the test accuracy of the model directly trained from the noisy data. (Higher is better.)

| Datasets | Noisy | DMLCBO | RMD-PCD | Approx | PDBO |
|---|---|---|---|---|---|
| FashionMNIST 8vs9 | $90.72 \pm 2.25$ | $\mathbf{96.63} \pm 0.69$ | $95.47 \pm 0.31$ | $95.70 \pm 0.26$ | $94.87 \pm 0.06$ |
| Gisette | $69.23 \pm 3.86$ | $\mathbf{86.17} \pm 1.02$ | $84.67 \pm 0.89$ | $86.00 \pm 1.66$ | $82.93 \pm 0.05$ |
| MNIST 1vs7 | $91.39 \pm 0.99$ | $\mathbf{93.45} \pm 1.58$ | $92.96 \pm 0.69$ | $92.94 \pm 0.69$ | $92.66 \pm 0.88$ |
| CodRNA | $77.65 \pm 0.27$ | $\mathbf{80.86} \pm 0.02$ | $79.56 \pm 0.18$ | $79.29 \pm 0.43$ | $80.52 \pm 0.08$ |

set may have different distributions. To reduce the discrepancy between the two distributions, each data point will be given an additional importance weight, which is called data hyper-clean. This problem can be formulated as

$$\min_{x \in \mathbb{R}^{d_1}} \sum_{\mathcal{D}_{val}} \ell\left(y^*(x)^\top a_i, b_i\right) \quad s.t. \quad y^*(x) = \arg\min_{\|y\|_1 \leq r} \sum_{\mathcal{D}_{tr}} [\sigma(x)]_i \ell\left(y^\top a_i, b_i\right),$$

where $r > 0$, $\mathcal{D}_{tr}$ and $\mathcal{D}_{val}$ denote the training set and validation set respectively; $a_i$ denotes the data instance; $b_i \in \{-1, +1\}$ denotes the label of $a_i$; $\sigma(\cdot) := 1/(1 + exp(-\cdot))$ is the Sigmoid function. In this experiment, an additional $\ell_1$ is added to the lower-level problem to ensure the sparsity of the model.

In this experiment, we evaluate all the methods on the datasets MNIST, FashionMNIST, CodRNA, and Gisette. For MNIST and FashionMNIST, we choose two classes to conduct a binary classification. In addition, we flip $30\%$ of the labels in the training set as the noisy data. We set $r = 1$ for all the datasets. For all the methods, we search the step size from the set $\{100, 10, 1, 10^{-1}, 10^{-2}, 10^{-3}\}$. Following the default setting in (Ji et al., 2021), we set $Q = 3$ and $\eta = 1$ for our method. In addition, we set $\eta_k = 1/\sqrt{100 + k}$, $c_1 = 9$ and $c_2 = 9$ for our method. For PDBO, RMD-PCD, and Approx, we set the inner iteration number at 3 for fair comparison. We run all the methods for 10000 iterations and evaluate the test accuracy for every 100 iteration.

**Training data poison attack.** In this experiment, we evaluate the performance of all the methods in training data poisoning. Assume we have pure training data $\mathcal{D}_{tr} = \{a_i, b_i\}_{i=1}^{N_{tr}}$ with several poisoned points $\mathcal{D}_{poi} = \{\hat{a}_j, \hat{b}_i\}_{j=1}^{N_{poi}}$ assigned the targeted labels. In this task, we search the poisoned data that can hurt the performance of the model on the targeted class. This problem can be formulated as

$$\max_{\|x\|_\infty \leq \epsilon'} \frac{1}{N_{val}^{tar}} \sum_{\mathcal{D}_{val}^{tar}} \ell(\theta(a_j; y^*(x)), b_j) \quad s.t. \quad y^*(x) = \arg\min_{\|y\| \leq r} \frac{1}{N} \sum_{\mathcal{D}_{tr} \bigcup \mathcal{D}_{poi}} \ell(\theta(a_i; y), b_i),$$

where $\epsilon' > 0$, $r > 0$, $N = N_{tr} + N_{poi}$ and $\mathcal{D}_{val}^{tar} = \{a_j, \hat{b}_j\}_{j=1}^{N_{val}^{tar}}$ denote the validation dataset with targeted labels.

In this experiment, we evaluate all the methods on the datasets MNIST and Cifar10. For MNIST, we set $\epsilon' = 0.1$ and $r = 10$ and we choose label 8 and 9 as the targeted label. For Cifar10, we set $\epsilon' = 0.1$ and $r = 10$ and we choose the label 4 and 6 as the targeted label. We use a network with two convolution layers and two fully-connected-layer layers for MNIST and a network with three

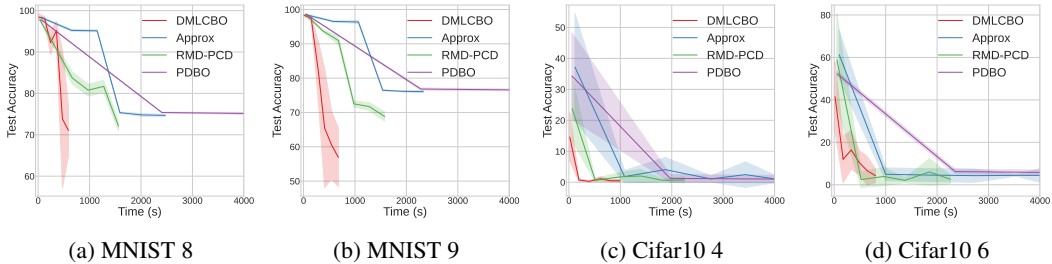

|   | (a) MNIST 8 | (b) MNIST 9 | (c) Cifar10 4 | (d) Cifar10 6 |

Figure 2: Test accuracy against training time of all the methods in training data poison attack .

Table 3: Test accuracy (%) with standard variance of the target class in training data poison attack. The second column shows the test accuracy of the model trained on the clean data. (Lower is better.)

| Datasets | Target | Train | DMLCBO | RMD-PCD | Approx | PDBO |
|---|---|---|---|---|---|---|
| MNIST | 8 | $98.17 \pm 0.15$ | $\mathbf{71.05} \pm 6.26$ | $72.09 \pm 1.17$ | $74.69 \pm 0.23$ | $74.71 \pm 0.56$ |
| MNIST | 9 | $98.07 \pm 0.15$ | $\mathbf{56.94} \pm 8.44$ | $68.81 \pm 1.36$ | $76.08 \pm 0.35$ | $76.14 \pm 0.19$ |
| Cifar10 | 4 | $68.43 \pm 1.29$ | $\mathbf{0.48} \pm 0.40$ | $0.74 \pm 0.96$ | $0.66 \pm 0.38$ | $0.80 \pm 1.10$ |
| Cifar10 | 6 | $77.08 \pm 5.52$ | $4.28 \pm 3.20$ | $\mathbf{2.56} \pm 2.09$ | $4.42 \pm 4.99$ | $3.50 \pm 1.10$ |

convolution layers and three fully-connected-layer layers for Cifar10, where the Relu function is used in each layer. For all the methods, we first train the model on clean data and use it as an initialization for the attack. Then, following the setting in (Mehra & Hamm, 2021; Shi & Gu, 2021), in each $x$ update iteration, we sample 200 data samples from $\mathcal{D}_{tr}$ and $\mathcal{D}_{poi}$ and 100 data samples from $\mathcal{D}_{val}^{tar}$ to perform all the methods. We run all the methods for 50 epochs. We search the step size from the set $\{10, 1, 10^{-1}, 10^{-2}\}$ for all the methods. Following the default setting in (Ji et al., 2021), we set $Q = 3$ and $\eta = 1$. In addition, we set $\eta_k = 1/\sqrt{100 + k}$, $c_1 = 9$ and $c_2 = 9$ for our method. For PDBO, RMD-PCD, and Approx, we set the inner iteration number at $T = 3$. We train a new model using the clean data and the poison data every epoch and evaluate the performance of the test data with the targeted loss.

### 5.3 RESULTS

We have presented the test accuracy results for all methods in Tables 2 and 3, and visualized the testing performance as a function of training time in Figures 1 and 2. Upon closer examination of the data presented in Tables 2 and 3, it becomes evident that our method consistently achieves superior results when compared to alternative approaches in both of these application scenarios. One possible reason is that using a small inner iteration number in RMD-PCD, Approx, and PDBO to solve the lower-level problem may lead to a model with poor performance which affects the hypergradient and the final performance of the model. Our DMLCBO sometimes has a large variance which is caused by the large variance of ZO estimation of the Jacobian of the projection operator. From all these results, we can conclude that our DMLCBO outperforms other methods in terms of accuracy and efficiency.

## 6 CONCLUSION

In this paper, we leverage Gaussian smoothing to design a hypergradient for the lower-level constrained BO problem. Then, using our new hypergradient, we propose a *single-loop single-timescale* algorithm based on the double-momentum method and adaptive step size method which updates the lower- and upper-level variables simultaneously. Theoretically, we prove our methods can converge to the stationary point with $\tilde{\mathcal{O}}(d_2^2 \epsilon^{-4})$. The experimental results in data hyper cleaning and poison training data attack demonstrate the efficiency and effectiveness of our proposed method.

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

APPENDIX FOR "DOUBLE MOMENTUM METHOD FOR LOWER-LEVEL CONSTRAINED BILEVEL OPTIMIZATION"

## A    ABLATION STUDY

Here we give the results of all the methods in hyper-clean with a larger iteration number to show the convergence performance. For our method, we set the iteration number at $100000$. The result are presented in Figurer 3.

In this section, we conduct ablation experiments on the hyper-parameters $Q$ and $\eta$. To control the variables, we explore the effect of each hyper-parameter while keeping the other hyper-parameters as default as shown in the experimental setups. We search the step size of both $x$ and $y$. We present the results in Tables 4, 5, 6, 7. We can find that increasing $Q$ will lead to a long training time. In addition, setting $Q = 1$ usually leads to the worst results, which is because setting $Q = 1$ means ignoring the inverse of the Hessian matrix in our hypergradient and may not converge to our stationary point.

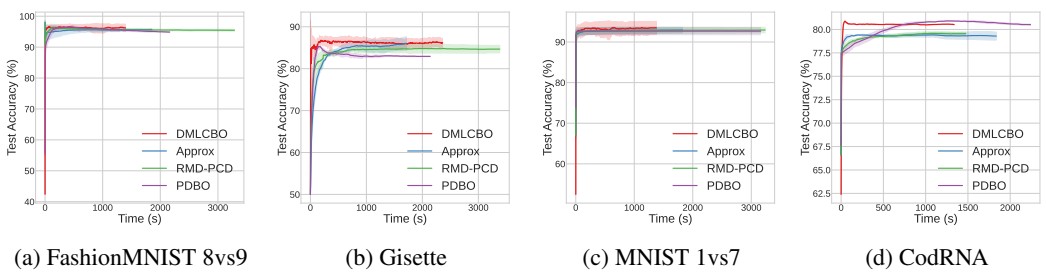

| (a) FashionMNIST 8vs9 | (b) Gisette | (c) MNIST 1vs7 | (d) CodRNA |
|---|---|---|---|

Figure 3: Test accuracy against training time of all the methods in data hyper-cleaning.

Table 4: Test accuracy (%) of our method with different Q in data hyper-cleaning. (Higher is better.)

| Datasets | Q=1 | Q=3 | Q=5 | Q=7 |
|---|---|---|---|---|
| FashionMNIST 8vs9 | $96.24 \pm 0.85$ | $\mathbf{96.63} \pm 0.69$ | $96.66 \pm 0.21$ | $96.45 \pm 0.36$ |
| Gisette | $86.12 \pm 0.33$ | $86.17 \pm 1.02$ | $86.45 \pm 0.64$ | $\mathbf{87.04} \pm 0.35$ |
| MNIST 1vs7 | $93.33 \pm 1.13$ | $93.45 \pm 1.58$ | $94.54 \pm 0.23$ | $\mathbf{94.25} \pm 0.25$ |
| CodRNA | $80.51 \pm 0.08$ | $\mathbf{80.86} \pm 0.02$ | $80.62 \pm 0.21$ | $81.87 \pm 2.37$ |

Table 5: Test accuracy (%) of our method with different $\eta$ in data hyper-cleaning. (Higher is better.)

| Datasets | $\eta = 1$ | $\eta = 0.1$ | $\eta = 0.01$ |
|---|---|---|---|
| FashionMNIST 8vs9 | $\mathbf{96.63} \pm 0.69$ | $96.24 \pm 0.17$ | $96.35 \pm 0.18$ |
| Gisette | $86.17 \pm 1.02$ | $86.05 \pm 1.36$ | $\mathbf{86.34} \pm 0.24$ |
| MNIST 1vs7 | $\mathbf{93.45} \pm 1.58$ | $93.25 \pm 0.87$ | $92.74 \pm 1.12$ |
| CodRNA | $\mathbf{80.86} \pm 0.02$ | $80.62 \pm 1.12$ | $80.55 \pm 0.25$ |

Table 6: Test accuracy (%) of our method with different $Q$ in training data poison attack. (lower is better.)

| Datasets | Target | $Q=1$ | $Q=3$ | $Q=4$ | $Q=5$ |
|----------|--------|-------|-------|-------|-------|
| MNIST | 8 | $71.60 \pm 7.11$ | $71.05 \pm 6.26$ | $71.15 \pm 3.35$ | $\mathbf{69.27} \pm 3.23$ |
| MNIST | 9 | $57.26 \pm 1.82$ | $\mathbf{56.94} \pm 8.44$ | $61.21 \pm 4.50$ | $58.28 \pm 3.73$ |
| Cifar10 | 4 | $2.30 \pm 0.69$ | $\mathbf{0.48} \pm 0.40$ | $0.73 \pm 0.54$ | $0.63 \pm 0.69$ |
| Cifar10 | 6 | $7.93 \pm 1.02$ | $4.28 \pm 3.20$ | $3.60 \pm 1.70$ | $\mathbf{0.55} \pm 0.05$ |

Table 7: Test accuracy (%) of our method with different $\eta$ in training data poison attack. (lower is better.)

| Datasets | Target | $\eta=1$ | $\eta=0.1$ | $\eta=0.01$ |
|----------|--------|----------|------------|-------------|
| MNIST | 8 | $\mathbf{71.05} \pm 6.26$ | $71.56 \pm 3.39$ | $71.22 \pm 1.41$ |
| MNIST | 9 | $\mathbf{56.94} \pm 8.44$ | $62.99 \pm 8.93$ | $62.56 \pm 3.56$ |
| Cifar10 | 4 | $\mathbf{0.48} \pm 0.40$ | $0.49 \pm 0.12$ | $0.48 \pm 0.21$ |
| Cifar10 | 6 | $\mathbf{4.28} \pm 3.20$ | $4.40 \pm 1.38$ | $4.38 \pm 0.75$ |

## B  PROOF OF PROPOSITION 1

*Proof.* According to Nesterov & Spokoiny (2017), for the first statement, since $\mathcal{P}_{\mathcal{Y}}(z^*)$ has Lipschitz continuous gradient, we have

$$
\|\nabla \mathcal{P}_{\mathcal{Y}\delta}(z^*) - \nabla \mathcal{P}_{\mathcal{Y}}(z^*)\|
$$

$$
\leq \frac{1}{\kappa \delta} \int \|\mathcal{P}_{\mathcal{Y}\delta}(z^* + \delta u) - \mathcal{P}_{\mathcal{Y}\delta}(z^*) - \delta \langle \nabla \mathcal{P}_{\mathcal{Y}\delta}(z^*), u \rangle\| \cdot \|u\| e^{-\frac{1}{2}\|u\|^2} du
$$

$$
\leq \frac{2\delta L}{\kappa} \int \|u\|^3 e^{-\frac{1}{2}\|u\|^2} du
$$

$$
\leq \frac{\delta L}{2}(d_2 + 3)^{3/2} \tag{12}
$$

In addition, since the projection operator is nonexpansive, we have

$$
\begin{aligned}
&\|\mathcal{P}_{\mathcal{Y}\delta}(z) - \mathcal{P}_{\mathcal{Y}\delta}(z')\| \\
&= \|\mathbb{E}_u[\mathcal{P}_{\mathcal{Y}}(z + \delta u) - \mathcal{P}_{\mathcal{Y}}(z' + \delta u)]\| \\
&\leq \mathbb{E}_u[\|\mathcal{P}_{\mathcal{Y}}(z + \delta u) - \mathcal{P}_{\mathcal{Y}}(z' + \delta u)\|] \\
&\leq \|z - z'\|
\end{aligned} \tag{13}
$$

In addition, we have

$$
\|\nabla \mathcal{P}_{\mathcal{Y}\delta}(z) - \nabla \mathcal{P}_{\mathcal{Y}\delta}(z')\|
$$

$$
\leq \frac{1}{\kappa \delta} \int \|\mathcal{P}_{\mathcal{Y}}(z + \delta u) - \mathcal{P}_{\mathcal{Y}}(z) + \mathcal{P}_{\mathcal{Y}}(z') - \mathcal{P}_{\mathcal{Y}}(z' + \delta u)\| \cdot \|u\| e^{-\frac{1}{2}\|u\|^2} du
$$

$$
\leq \frac{1}{\kappa \delta} \int \left( \|\mathcal{P}_{\mathcal{Y}}(z + \delta u) - \mathcal{P}_{\mathcal{Y}}(z' + \delta u)\| + \|\mathcal{P}_{\mathcal{Y}}(z') - \mathcal{P}_{\mathcal{Y}}(z)\| \right) \|u\| e^{-\frac{1}{2}\|u\|^2} du
$$

$$
\leq \frac{2}{\kappa \delta} \|z - z'\| \int \|u\| e^{-\frac{1}{2}\|u\|^2} du
$$

$$
= \frac{2\sqrt{d_2}}{\delta} \|z - z'\| \tag{14}
$$

$\square$

## C    PROOF OF LEMMA 3

*Proof.* From Nesterov & Spokoiny (2017), we can easily obtain $\mathbb{E}_u \left[ \bar{H}(z;u) \right] = \nabla \mathcal{P}_{\mathcal{Y}\delta}(z)$. In addition, we have

$$
\begin{aligned}
&\mathbb{E}_u \left[ \| \bar{H}(z;u) \|^2 \right] \\
=&\frac{1}{\delta^2} \mathbb{E}_u \left[ \| \mathcal{P}_{\mathcal{Y}}(z + \delta u) - \mathcal{P}_{\mathcal{Y}}(z) \|^2 \| u \|^2 \right] \\
\leq&\mathbb{E}_u \left[ \| u \|^4 \right] \\
\leq&(d_2 + 4)^2
\end{aligned}
\tag{15}
$$

Then, we can easily obtain

$$
\mathbb{E}_u \left[ \left\| \bar{H}(z;u) - \nabla \mathcal{P}_{\mathcal{Y}\delta}(z) \right\|^2 \right] \leq 2\mathbb{E}_u \left[ \left\| \bar{H}(z;u) \right\|^2 \right] + 2\mathbb{E}_u \left[ \left\| \nabla \mathcal{P}_{\mathcal{Y}\delta}(z) \right\|^2 \right] \leq (d_2 + 4)^2 + 2
\tag{16}
$$

$\square$

## D    PROOF OF LEMMA 4

*Proof.* For convenience, define $\bar{G}_{yy} = Q \prod_{i=1}^{c(Q)} \left( (I_{d_2} - \eta \nabla_{yy}^2 g(x,y)) \bar{H}(z;u^i)^\top \right)$ and $G_{yy} = \left[ I_{d_2} - (I_{d_2} - \eta \nabla_{yy}^2 g(x,y)) \nabla \mathcal{P}_{\mathcal{Y}\delta}(z)^\top \right]^{-1}$. We set $\eta \leq \dfrac{1}{L_g}$ We have

$$
\mathbb{E}_{\bar{\xi}}[\bar{\nabla} f_\delta(x,y;\bar{\xi})] = \nabla_x f(x,y) - \eta \nabla_{xy}^2 g(x,y) \nabla \mathcal{P}_{\mathcal{Y}\delta}(z)^\top \mathbb{E} \left[ \bar{G}_{yy} \right] \nabla_y f(x,y)
\tag{17}
$$

We have

$$
\begin{aligned}
&\left\| \nabla f_\delta(x,y) - \mathbb{E}[\bar{\nabla} f_\delta(x,y;\bar{\xi})] \right\| \\
=&\left\| \eta \nabla_{xy}^2 g(x,y) \partial_z \mathcal{P}_{\mathcal{Y}\delta}(z)^\top \left\{ \mathbb{E} \left[ \bar{G}_{yy} \right] - G_{yy} \right\} \nabla_y f(x,y) \right\| \\
\leq&\eta C_{gxy} C_{fy} \left\| \mathbb{E} \left[ \bar{G}_{yy} \right] - G_{yy} \right\|
\end{aligned}
\tag{18}
$$

where the third inequality is due to the non-expansive of the projector operation.

Due to the independency of $u, c(Q)$, we have

$$
\begin{aligned}
&\mathbb{E} \left[ \bar{G}_{yy} \right] \\
=&\mathbb{E} \left[ Q \prod_{i=1}^{c(Q)} \left( (I_{d_2} - \eta \nabla_{yy}^2 g(x,y)) \bar{H}(z;u^i)^\top \right) \right] \\
=&Q \mathbb{E}_{c(Q)} \left[ \mathbb{E}_u \left[ \prod_{i=1}^{c(Q)} \left( (I_{d_2} - \eta \nabla_{yy}^2 g(x,y)) \bar{H}(z;u^i)^\top \right) \right] \right] \\
=&Q \mathbb{E}_{c(Q)} \left[ (I_{d_2} - \eta \nabla_{yy}^2 g(x,y)) \nabla \mathcal{P}_{\mathcal{Y}\delta}(z)^\top \right]^{c(Q)} \\
=&\sum_{i=0}^{Q-1} \left[ (I_{d_2} - \eta \nabla_{yy}^2 g(x,y)) \nabla \mathcal{P}_{\mathcal{Y}\delta}(z)^\top \right]^i
\end{aligned}
\tag{19}
$$

In addition, we have

$$
\begin{aligned}
&G_{yy} \\
=&\left[ I_{d_2} - (I_{d_2} - \eta \nabla_{yy}^2 g(x,y)) \nabla \mathcal{P}_{\mathcal{Y}\delta}(z)^\top \right]^{-1} \\
=&\sum_{i=0}^{\infty} \left[ (I_{d_2} - \eta \nabla_{yy}^2 g(x,y)) \nabla \mathcal{P}_{\mathcal{Y}\delta}(z)^\top \right]^i \\
=&\mathbb{E} \left[ Q \prod_{i=1}^{c(Q)} \left( (I_{d_2} - \eta \nabla_{yy}^2 g(x,y)) \bar{H}(z;u^i)^\top \right) \right] + \sum_{i=Q}^{\infty} \left[ (I_{d_2} - \eta \nabla_{yy}^2 g(x,y)) \nabla \mathcal{P}_{\mathcal{Y}\delta}(z)^\top \right]^i
\end{aligned}
\tag{20}
$$

where $z = y - \eta\nabla_y g(x,y)$, which implies that

$$
\begin{aligned}
&\left\|\mathbb{E}\left[\bar{G}_{yy}\right] - \bar{G}_{yy}\right\| \\
=&\left\|\left[(I_{d_2} - (I_{d_2} - \eta\nabla_{yy}^2 g(x,y))\nabla\mathcal{P}_{y\delta}(z)^\top\right]^{-1} - \mathbb{E}\left[Q\prod_{i=1}^{c(Q)}\left((I_{d_2} - \eta\nabla_{yy}^2 g(x,y;\zeta^i))\bar{H}(z;u^i)^\top\right)\right]\right\| \\
\leq&\sum_{i=Q}^{\infty}\left\|(I_{d_2} - \eta\nabla_{yy}^2 g(x,y))\nabla\mathcal{P}_{y\delta}(z)^\top\right\|^i \\
\leq&\sum_{i=Q}^{\infty}\left\|I_{d_2} - \eta\nabla_{yy}^2 g(x,y))\right\|^i\left\|\nabla\mathcal{P}_{y\delta}(z)\right\|^i \\
\leq&\frac{1}{\eta\mu_g}(1 - \eta\mu_g)^Q
\end{aligned}
\tag{21}
$$

Thus, we have

$$
\left\|\nabla f_\delta(x,y) - \mathbb{E}[\bar{\nabla}f_\delta(x,y;\bar{\xi})]\right\| \leq \frac{C_{gxy}C_{fy}}{\mu_g}(1 - \eta\mu_g)^Q
\tag{22}
$$

Then, we prove the bound on the variance. we have

$$
\begin{aligned}
&\mathbb{E}\left[\left\|\bar{\nabla}f_\delta(x,y;\bar{\xi}) - \mathbb{E}\left[\bar{\nabla}f_\delta(x,y;\bar{\xi})\right]\right\|^2\right] \\
=&\mathbb{E}\left[\left\|\eta\nabla_{xy}^2 g(x,y)\bar{H}(z;u^0)^\top\bar{G}_{yy}\nabla_y f(x,y) - \eta\nabla_{xy}^2 g(x,y)\nabla\mathcal{P}_{y\delta}(z)^\top\mathbb{E}[\bar{G}_{yy}]\nabla_y f(x,y)\right\|^2\right] \\
\leq&2\eta^2\left\|\nabla_{xy}^2 g(x,y)\right\|^2\mathbb{E}\left[\left\|\bar{H}(z;u^0) - \nabla\mathcal{P}_{y\delta}(z)\right\|^2\right]\mathbb{E}\left[\left\|\bar{G}_{yy}\right\|^2\right]\left\|\nabla_y f(x,y)\right\|^2 \\
&+ 2\eta^2\left\|\nabla_{xy}g(x,y)\right\|^2\left\|\nabla\mathcal{P}_{y\delta}(z)\right\|^2\mathbb{E}\left[\left\|\bar{G}_{yy} - \mathbb{E}[\bar{G}_{yy}]\right\|^2\right]\left\|\nabla_y f(x,y)\right\|^2
\end{aligned}
\tag{23}
$$

For the first term in the above inequality, we have

$$
\mathbb{E}\left[\left\|\bar{H}(z;u^0) - \nabla\mathcal{P}_{y\delta}(z)\right\|^2\right] \leq (d_2 + 4)^2 + 2
\tag{24}
$$

For $\mathbb{E}[\|\bar{G}_{yy}\|^2]$, we have

$$
\begin{aligned}
\mathbb{E}[\|\bar{G}_{yy}\|^2] =&\sum_{q=0}^{Q-1}\mathbb{E}\left[\left\|\prod_{i=1}^{q}\left((I_{d_2} - \eta\nabla_{yy}^2 g(x,y))\bar{H}(z;u^i)^\top\right)\right\|^2\right] \\
\leq&\sum_{q=0}^{Q-1}((1 - \eta\mu_g))^q \\
\leq&\frac{1}{1 - (1 - \eta\mu_g)(d_2 + 4)}
\end{aligned}
\tag{25}
$$

where $C = d_2 + 4$ is the bound of $\mathbb{E}[\|\bar{H}(z;u^i)\|]$ given in Nesterov & Spokoiny (2017) and the last inequality is obtained by setting $\frac{1}{\mu_g}(1 - \frac{1}{d_2 + 4}) \leq \eta \leq \frac{1}{\mu_g}$.

We can also derive that

$$\left\| \mathbb{E}\left[\bar{G}_{yy}\right]\right\|$$
$$=\| \sum_{i=0}^{Q-1} \left[(I_{d_2} - \eta\nabla^2_{yy}g(x,y))\nabla\mathcal{P}_{\mathcal{Y}\delta}(z)^\top\right]^i \|$$
$$\leq \sum_{i=0}^{Q-1} \|(I_{d_2} - \eta\nabla^2_{yy}g(x,y))\nabla\mathcal{P}_{\mathcal{Y}\delta}(z)^\top\|^i$$
$$\leq \sum_{i=0}^{Q-1} (1-\eta\mu_g)^i$$
$$\leq \frac{1}{\eta\mu_g} \tag{26}$$

Then, we have

$$\mathbb{E}\left[\|G_{yy} - \mathbb{E}[G_{yy}]\|^2\right] \leq \frac{2}{1-(1-\eta\mu_g)(d_2+4)} + \frac{2}{\eta^2\mu_g^2} \tag{27}$$

Therefore, combining the above inequalities, we can bound the variance as follows,

$$\mathbb{E}\left[\left\|\bar{\nabla}f(x,y;\bar{\xi}) - \mathbb{E}\left[\bar{\nabla}f(x,y;\bar{\xi})\right]\right\|^2\right]$$
$$\leq 2\eta^2 C_{gxy}^2((d_2+4)^2+2)C_{fy}^2\frac{1}{1-(1-\eta\mu_g)(d_2+4)}$$
$$+ 2\eta^2 C_{gxy}^2(\frac{2}{1-(1-\eta\mu_g)(d_2+4)} + \frac{2}{\eta^2\mu_g^2})C_{fy}^2 \tag{28}$$

That completes the proof. $\square$

## E    ROUTE MAP OF OUR CONVERGENCE ANALYSIS

Here we give a simple route map of our convergence analysis.

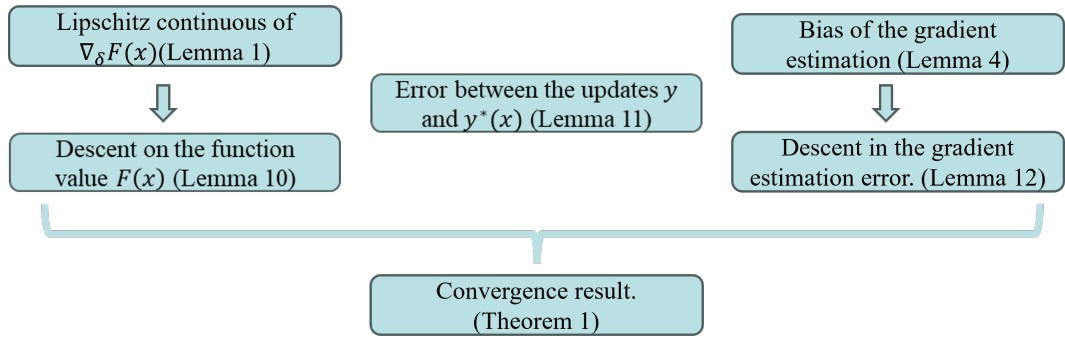

Figure 4: Route map of convergence analysis.

## F    LIPSCHITZ CONTINUOUSNESS OF $\nabla F_\delta(x)$

Here, we prove $\nabla F_\delta(x)$ is Lipschitz continuous. We first give several useful lemmas.

**Lemma 6.** *(**Lipschitz continuous of the optimal solution to the lower-level problem.**)  Under Assumptions 2, we have*

$$\|y^*(x_1) - y^*(x_2)\| \leq L_y\|x_1 - x_2\| \tag{29}$$

*where $L_y = \dfrac{L_g}{\mu_g}$.*

*Proof.* Here, we follow the proof in Van Leeuwen & Aravkin (2021). For given $x_1$ and $x_2$, we have the corresponding unique optimal solutions $y^*(x_1)$ and $y^*(x_2)$. For the constrained lower-level problem, we have the following optimal conditions

$$0 \in \nabla_y g(x_1, y^*(x_1)) + \partial \delta_{\mathcal{Y}}(y^*(x_1)), \quad 0 \in \nabla_y g(x_2, y^*(x_2)) + \partial \delta_{\mathcal{Y}}(y^*(x_2)) \tag{30}$$

where $\delta_Y(\cdot)$ is the identify function of the constriant.

For any given $\tilde{x}$, we have

$$\mu_g \|y^*(x_1) - y^*(x_2)\|^2 \le \langle (\nabla_y g(\tilde{x}, y^*(x_1)) + h_1) - (\nabla_y g(\tilde{x}, y^*(x_2)) + h_2), y^*(x_1) - y^*(x_2) \rangle \tag{31}$$

where $h_1 \in \partial \delta_{\mathcal{Y}}(y^*(x_1))$ and $h_2 \in \partial \delta_{\mathcal{Y}}(y^*(x_2))$. Make the particular choices $h_1 = -\nabla_y g(x_1, y^*(x_1)) \in \partial \delta_{\mathcal{Y}}(y^*(x_1))$ and $h_2 = -\nabla_y g(x_2, y^*(x_2)) \in \partial \delta_{\mathcal{Y}}(y^*(x_2))$, we have

$$
\begin{aligned}
&\mu_g \|y^*(x_1) - y^*(x_2)\|^2 \\
&\le \langle \nabla_y g(\tilde{x}, y^*(x_1)) - \nabla_y g(x_1, y^*(x_1)), y^*(x_1) - y^*(x_2) \rangle \\
&\quad + \langle \nabla_y g(x_2, y^*(x_2)) - \nabla_y g(\tilde{x}, y^*(x_2)), y^*(x_1) - y^*(x_2) \rangle
\end{aligned} \tag{32}
$$

Then, setting $\tilde{x} = x_1$, we have

$$
\begin{aligned}
&\mu_g \|y^*(x_1) - y^*(x_2)\|^2 \\
&\le \langle \nabla_y g(x_2, y^*(x_2)) - \nabla_y g(x_1, y^*(x_2)), y^*(x_1) - y^*(x_2) \rangle \\
&\le \|\nabla_y g(x_2, y^*(x_2)) - \nabla_y g(x_1, y^*(x_2))\| \|y^*(x_1) - y^*(x_2)\| \\
&\le L_g \|x_1 - x_2\| \|y^*(x_1) - y^*(x_2)\|
\end{aligned} \tag{33}
$$

where the last inequality is due to Assumption 2. Rearrange above inequality, we have

$$\|y^*(x_1) - y^*(x_2)\| \le \frac{L_g}{\mu_g} \|x_1 - x_2\| \tag{34}$$

That completes the proof. $\qquad\square$

**Lemma 7.** *(**Lipschitz continuous of the approximation of hypergradient on $x$ and $y$.**) Under Assumptions 1, 2, $\nabla f_\delta(x, y)$ is Lipschitz continuous on $y \in \mathcal{Y}$ and $x \in \mathcal{X}$, respectively, such that we have*

$$\|\nabla f_\delta(x, y_1) - \nabla f_\delta(x, y_2)\| \le L_1(\frac{\sqrt{d_2}}{\delta}) \|y_1 - y_2\| \tag{35}$$

$$\|\nabla f_\delta(x_1, y) - \nabla f_\delta(x_2, y)\| \le L_2(\frac{\sqrt{d_2}}{\delta}) \|x_1 - x_2\| \tag{36}$$

*where* $L_1(\frac{\sqrt{d_2}}{\delta}) = L_f + \frac{L_{gxy} C_{fy}}{\mu_g} + \frac{C_{gxy} C_{fy}}{\mu_g} (\frac{2\sqrt{d_2}}{\delta} + \eta \frac{2\sqrt{d_2}}{\delta} L_g) + C_{gxy} C_{fy} \frac{1}{\eta \mu_g^2} (\frac{2\sqrt{d_2}}{\delta} + \eta \frac{2\sqrt{d_2}}{\delta} L_g)(1 - \eta \mu_g) + \frac{L_{gyy}}{\mu_g^2} + \frac{C_{gxy}}{\mu_g} L_f$ *and* $L_2(\frac{\sqrt{d_2}}{\delta}) = L_f + \frac{L_{gxy} C_{fy}}{\mu_g} + \frac{C_{gxy} C_{fy}}{\mu_g} \eta \frac{2\sqrt{d_2}}{\delta} L_g + C_{gxy} C_{fy} \left( \eta \frac{1}{\mu_g^2} \frac{2\sqrt{d_2}}{\delta} L_g (1 - \eta \mu_g) + \frac{L_{gyy}}{\mu_g^2} \right) + \frac{C_{gxy}}{\mu_g} L_f$

*Proof.* Using the definition of $\nabla f_\delta(x, y)$, $z_1 = y_1 - \nabla_y g(x, y_1)$ and $z_2 = y_2 - \nabla_y g(x, y_2)$, we have

$\|\nabla f_\delta(x, y_1) - \nabla f_\delta(x, y_2)\|$

$=\|\nabla_x f(x, y_1) - \eta \nabla^2_{xy} g(x, y_1) \nabla \mathcal{P}_{\mathcal{Y}\delta}(z_1)^\top \left[(I_{d_2} - (I_{d_2} - \eta \nabla^2_{yy} g(x, y_1)) \nabla \mathcal{P}_{\mathcal{Y}\delta}(z_1)^\top\right]^{-1} \nabla_y f(x, y_1)$

$\quad - \nabla_x f(x, y_2) + \eta \nabla^2_{xy} g(x, y_2) \nabla \mathcal{P}_{\mathcal{Y}\delta}(z_2)^\top \left[(I_{d_2} - (I_{d_2} - \eta \nabla^2_{yy} g(x, y_2)) \nabla \mathcal{P}_{\mathcal{Y}\delta}(z_2)^\top\right]^{-1} \nabla_y f(x, y_2)\|$

$\leq \|\nabla_x f(x, y_1) - \nabla f(x, y_2)\|$

$\quad + \eta \|\nabla^2_{xy} g(x, y_2) - \nabla^2_{xy} g(x, y_1)\| \|\nabla \mathcal{P}_{\mathcal{Y}\delta}(z_2)\| \| \left[I_{d_2} - (I_{d_2} - \eta \nabla^2_{yy} g(x, y_2)) \nabla \mathcal{P}_{\mathcal{Y}\delta}(z_2)^\top\right]^{-1} \| \|\nabla_y f(x, y_2)\|$

$\quad + \eta \|\nabla^2_{xy} g(x, y_1)\| \|\nabla \mathcal{P}_{\mathcal{Y}\delta}(z_2) - \nabla \mathcal{P}_{\mathcal{Y}\delta}(z_1)\| \| \left[I_{d_2} - (I_{d_2} - \eta \nabla^2_{yy} g(x, y_2)) \nabla \mathcal{P}_{\mathcal{Y}\delta}(z_2)^\top\right]^{-1} \| \|\nabla_y f(x, y_2)\|$

$\quad + \eta \|\nabla^2_{xy} g(x, y_1)\| \|\nabla \mathcal{P}_{\mathcal{Y}\delta}(z_1)\| \| \left[I_{d_2} - (I_{d_2} - \eta \nabla^2_{yy} g(x, y_2)) \nabla \mathcal{P}_{\mathcal{Y}\delta}(z_2)^\top\right]^{-1}$

$\quad - \left[I_{d_2} - (I_{d_2} - \eta \nabla^2_{yy} g(x, y_1)) \nabla \mathcal{P}_{\mathcal{Y}\delta}(z_1)^\top\right]^{-1} \| \|\nabla_y f(x, y_2)\|$

$\quad + \eta \|\nabla^2_{xy} g(x, y_1)\| \|\nabla \mathcal{P}_{\mathcal{Y}\delta}(z_1)\| \| \left[I_{d_2} - (I_{d_2} - \eta \nabla^2_{yy} g(x, y_1)) \nabla \mathcal{P}_{\mathcal{Y}\delta}(z_1)^\top\right]^{-1} \| \|\nabla_y f(x, y_2) - \nabla_y f(x, y_1)\|$

$$\tag{37}$$

We have

$$\|\nabla_x f(x, y_1) - \nabla_x f(x, y_2)\| \leq L_f \|y_1 - y_2\|, \tag{38}$$

$$\|\nabla_y f(x, y_1) - \nabla_y f(x, y_2)\| \leq L_f \|y_1 - y_2\|, \tag{39}$$

$$\|\nabla^2_{xy} g(x, y_2) - \nabla^2_{xy} g(x, y_1)\| \leq L_{gxy} \|y_2 - y_1\|, \tag{40}$$

$$\|\nabla \mathcal{P}_{\mathcal{Y}\delta}(z_1)\| \leq 1, \quad \|\nabla f(x, y_2)\| \leq C_{fy}, \quad \|\nabla^2_{xy} g(x, y_1)\| \leq C_{gxy} \tag{41}$$

Since $\|(I_{d_2} - \eta \nabla^2_{yy} g(x, y_2)) \nabla \mathcal{P}_{\mathcal{Y}\delta}(z_2)^\top\| \leq \|\nabla \mathcal{P}_{\mathcal{Y}\delta}(z_2)\| \|I_{d_2} - \eta \nabla^2_{yy} g(x, y_2)\| \leq 1 - \eta \mu_g \leq 1$
we have

$$\| \left[I_{d_2} - (I_{d_2} - \eta \nabla^2_{yy} g(x, y_2)) \nabla \mathcal{P}_{\mathcal{Y}\delta}(z_2)^\top\right]^{-1} \|$$

$$\leq \frac{1}{1 - \|(I_{d_2} - \eta \nabla^2_{yy} g(x, y_2)) \nabla \mathcal{P}_{\mathcal{Y}\delta}(z_2)^\top\|}$$

$$\leq \frac{1}{\eta \mu_g} \tag{42}$$

$$\|\nabla \mathcal{P}_{\mathcal{Y}\delta}(z_2) - \nabla \mathcal{P}_{\mathcal{Y}\delta}(z_1)\|$$

$$\leq \frac{2\sqrt{d_2}}{\delta} \|z_2 - z_1\|$$

$$= \frac{2\sqrt{d_2}}{\delta} \|y_2 - \eta \nabla_y g(x, y_2) - y_1 + \eta \nabla_y g(x, y_1)\|$$

$$\leq \frac{2\sqrt{d_2}}{\delta} \|y_2 - y_1\| + \eta \frac{2\sqrt{d_2}}{\delta} \|\nabla_y g(x, y_1) - \eta \nabla_y g(x, y_2)\|$$

$$\leq (\frac{2\sqrt{d_2}}{\delta} + \eta \frac{2\sqrt{d_2}}{\delta} L_g) \|y_2 - y_1\| \tag{43}$$

Using the inequality $\|H_2^{-1} - H_1^{-1}\| \leq \|H_1^{-1}(H_1 - H_2)H_2^{-1}\| \leq \|H_1^{-1}\| \|H_1 - H_2\| \|H_2^{-1}\|$, we have

$$\| \left[I_{d_2} - (I_{d_2} - \eta \nabla^2_{yy} g(x, y_2)) \nabla \mathcal{P}_{\mathcal{Y}\delta}(z_2)^\top\right]^{-1} - \left[I_{d_2} - (I_{d_2} - \eta \nabla^2_{yy} g(x, y_1)) \nabla \mathcal{P}_{\mathcal{Y}\delta}(z_1)^\top\right]^{-1} \|$$

$$\leq \frac{1}{\eta^2 \mu_g^2} \|(I_{d_2} - \eta \nabla^2_{yy} g(x, y_2)) \nabla \mathcal{P}_{\mathcal{Y}\delta}(z_2)^\top - (I_{d_2} - \eta \nabla^2_{yy} g(x, y_1)) \nabla \mathcal{P}_{\mathcal{Y}\delta}(z_1)^\top\|$$

$$\leq \frac{1}{\eta^2 \mu_g^2} \|\nabla \mathcal{P}_{\mathcal{Y}\delta}(z_2) - \nabla \mathcal{P}_{\mathcal{Y}\delta}(z_1)\| \|I_{d_2} - \eta \nabla^2_{yy} g(x, y_2)\|$$

$$\quad + \frac{1}{\eta^2 \mu_g^2} \|\nabla \mathcal{P}_{\mathcal{Y}\delta}(z_1)\| \|I_{d_2} - \eta \nabla^2_{yy} g(x, y_2) - I_{d_2} + \eta \nabla^2_{yy} g(x, y_1)\|$$

$$\leq \left( \frac{1}{\eta^2 \mu_g^2} (\frac{2\sqrt{d_2}}{\delta} + \eta \frac{2\sqrt{d_2}}{\delta} L_g)(1 - \eta \mu_g) + \frac{L_{gyy}}{\eta \mu_g^2} \right) \|y_2 - y_1\| \tag{44}$$

Therefore, using the above inequalities, we obtain

$$
\begin{aligned}
&\|\nabla f_\delta(x, y_1) - \nabla f_\delta(x, y_2 \nabla f_\delta(x, y))\| \\
\leq& L_f + \frac{L_{gxy} C_{fy}}{\mu_g} + \frac{C_{gxy} C_{fy}}{\mu_g}(\frac{2\sqrt{d_2}}{\delta} + \eta \frac{2\sqrt{d_2}}{\delta} L_g) \\
&+ C_{gxy} C_{fy} \left( \frac{1}{\eta \mu_g^2}(\frac{2\sqrt{d_2}}{\delta} + \eta \frac{2\sqrt{d_2}}{\delta} L_g)(1 - \eta\mu_g) + \frac{L_{gyy}}{\mu_g^2} \right) + \frac{C_{gxy}}{\mu_g} L_f \|y_2 - y_1\|
\end{aligned}
\tag{45}
$$

For the second statement, let $z_1 = y - \nabla_y g(x_1, y)$ and $z_2 = y - \nabla_y g(x_2, y)$, we have

$$
\begin{aligned}
&\|\nabla f_\delta(x_1, y) - \nabla f_\delta(x_2, y)\| \\
=&\|\nabla_x f(x_1, y) - \eta \nabla_{xy}^2 g(x_1, y) \nabla \mathcal{P}_{\mathcal{Y}\delta}(z_1)^\top \left[ I_{d_2} - (I_{d_2} - \eta\nabla_{yy}^2 g(x_1, y)) \nabla \mathcal{P}_{\mathcal{Y}\delta}(z_1)^\top \right]^{-1} \nabla_y f(x_1, y) \\
&\quad - \nabla_x f(x_2, y) + \eta \nabla_{xy}^2 g(x_2, y) \nabla \mathcal{P}_{\mathcal{Y}\delta}(z_2)^\top \left[ I_{d_2} - (I_{d_2} - \eta\nabla_{yy}^2 g(x_2, y)) \nabla \mathcal{P}_{\mathcal{Y}\delta}(z_2)^\top \right]^{-1} \nabla_y f(x_2, y)\| \\
\leq&\|\nabla_x f(x_1, y) - \nabla f(x_2, y)\| \\
&\quad + \eta \|\nabla_{xy}^2 g(x_2, y) - \nabla_{xy}^2 g(x_1, y)\| \|\nabla \mathcal{P}_{\mathcal{Y}\delta}(z_2)\| \| \left[ I_{d_2} - (I_{d_2} - \eta\nabla_{yy}^2 g(x_2, y)) \nabla \mathcal{P}_{\mathcal{Y}\delta}(z_2)^\top \right]^{-1} \| \|\nabla_y f(x_2, y)\| \\
&\quad + \eta \|\nabla_{xy}^2 g(x_1, y)\| \|\nabla \mathcal{P}_{\mathcal{Y}\delta}(z_2) - \nabla \mathcal{P}_{\mathcal{Y}\delta}(z_1)\| \| \left[ I_{d_2} - (I_{d_2} - \eta\nabla_{yy}^2 g(x_2, y)) \nabla \mathcal{P}_{\mathcal{Y}\delta}(z_2)^\top \right]^{-1} \| \|\nabla_y f(x_2, y)\| \\
&\quad + \eta \|\nabla_{xy}^2 g(x_1, y)\| \|\nabla \mathcal{P}_{\mathcal{Y}\delta}(z_1)\| \| \left[ I_{d_2} - (I_{d_2} - \eta\nabla_{yy}^2 g(x_2, y)) \nabla \mathcal{P}_{\mathcal{Y}\delta}(z_2)^\top \right]^{-1} \\
&\quad - \left[ I_{d_2} - (I_{d_2} - \eta\nabla_{yy}^2 g(x_1, y)) \nabla \mathcal{P}_{\mathcal{Y}\delta}(z_1)^\top \right]^{-1} \| \|\nabla_y f(x_2, y)\| \\
&\quad + \eta \|\nabla_{xy}^2 g(x_1, y)\| \|\nabla \mathcal{P}_{\mathcal{Y}\delta}(z_1)\| \| \left[ I_{d_2} - (I_{d_2} - \eta\nabla_{yy}^2 g(x_1, y)) \nabla \mathcal{P}_{\mathcal{Y}\delta}(z_1)^\top \right]^{-1} \| \|\nabla_y f(x_2, y) - \nabla_y f(x_1, y)\|
\end{aligned}
\tag{46}
$$

We have

$$
\|\nabla_x f(x_1, y) - \nabla_x f(x_2, y)\| \leq L_f \|x_2 - x_1\|
\tag{47}
$$

$$
\|\nabla_y f(x_1, y) - \nabla_y f(x_2, y)\| \leq L_f \|x_2 - x_1\|
\tag{48}
$$

$$
\|\nabla_{xy}^2 g(x_2, y) - \nabla_{xy}^2 g(x_1, y)\| \leq L_{gxy} \|x_2 - x_1\|
\tag{49}
$$

$$
\begin{aligned}
&\|\nabla \mathcal{P}_{\mathcal{Y}\delta}(z_2) - \nabla \mathcal{P}_{\mathcal{Y}\delta}(z_1)\| \\
\leq& \frac{2\sqrt{d_2}}{\delta} \|z_2 - z_1\| \\
=& \frac{2\sqrt{d_2}}{\delta} \|y - \eta \nabla_y g(x_2, y) - y + \eta \nabla_y g(x_1, y)\| \\
\leq& \eta \frac{2\sqrt{d_2}}{\delta} \|\nabla_y g(x_1, y) - \eta \nabla_y g(x_2, y)\| \\
\leq& \eta \frac{2\sqrt{d_2}}{\delta} L_g \|x_1 - x_2\|
\end{aligned}
\tag{50}
$$

$$
\begin{aligned}
&\| \left[ I_{d_2} - (I_{d_2} - \eta\nabla_{yy}^2 g(x_2, y)) \nabla \mathcal{P}_{\mathcal{Y}\delta}(z_2)^\top \right]^{-1} - \left[ I_{d_2} - (I_{d_2} - \eta\nabla_{yy}^2 g(x_1, y)) \nabla \mathcal{P}_{\mathcal{Y}\delta}(z_1)^\top \right]^{-1} \| \\
\leq& \frac{1}{\eta^2 \mu_g^2} \|(I_{d_2} - \eta\nabla_{yy}^2 g(x_1, y)) \nabla \mathcal{P}_{\mathcal{Y}\delta}(z_1)^\top - (I_{d_2} - \eta\nabla_{yy}^2 g(x_2, y)) \nabla \mathcal{P}_{\mathcal{Y}\delta}(z_2)^\top \| \\
\leq& \frac{1}{\eta^2 \mu_g^2} \|\nabla \mathcal{P}_{\mathcal{Y}\delta}(z_1) - \nabla \mathcal{P}_{\mathcal{Y}\delta}(z_2)\| \|I_{d_2} - \eta\nabla_{yy}^2 g(x_1, y)\| \\
&\quad + \frac{1}{\eta^2 \mu_g^2} \|\nabla \mathcal{P}_{\mathcal{Y}\delta}(z_2)\| \|I_{d_2} - \eta\nabla_{yy}^2 g(x_1, y) - I_{d_2} + \eta\nabla_{yy}^2 g(x_2, y)\| \\
\leq& \left( \frac{1}{\eta \mu_g^2} \frac{2\sqrt{d_2}}{\delta} L_g (1 - \eta\mu_g) + \frac{L_{gyy}}{\eta \mu_g^2} \right) \|x_1 - x_2\|
\end{aligned}
\tag{51}
$$

Therefore, we have

$$\|\nabla f_\delta(x_1, y) - \nabla f_\delta(x_2, y)\|$$

$$\leq (L_f + \frac{L_{gxy}C_{fy}}{\mu_g} + \frac{C_{gxy}C_{fy}}{\mu_g}\eta\frac{2\sqrt{d_2}}{\delta}L_g + C_{gxy}C_{fy}\left(\eta\frac{1}{\mu_g^2}\frac{2\sqrt{d_2}}{\delta}L_g(1-\eta\mu_g) + \frac{L_{gyy}}{\mu_g^2}\right)$$

$$+ \frac{C_{gxy}}{\mu_g}L_f)\|x_1 - x_2\| \tag{52}$$

$$\square$$

## F.1 Proof of Lemma 1

Here we first give a detailed version of Lemma 1 and then give the proof.

**Lemma 8.** *(Lipschitz continous of $\nabla F_\delta(x)$.) Under Assumptions 1, 2 and Lemma 3, we have $\nabla F_\delta(x)$ is Lipschitz continuous w.r.t $x$, such that*

$$\|\nabla F_\delta(x_1) - \nabla F_\delta(x_2)\| \leq L_{F_\delta}(\frac{\sqrt{d_2}}{\delta})\|x_1 - x_2\| \tag{53}$$

*where $L_{F_\delta}(\frac{\sqrt{d_2}}{\delta}) = \frac{L_g}{\mu_g}L_1(\frac{\sqrt{d_2}}{\delta}) + L_2(\frac{\sqrt{d_2}}{\delta})$, $L_1(\frac{\sqrt{d_2}}{\delta}) = L_f + \frac{L_{gxy}C_{fy}}{\mu_g} + \frac{C_{gxy}C_{fy}}{\mu_g}(\frac{2\sqrt{d_2}}{\delta} + \eta\frac{2\sqrt{d_2}}{\delta}L_g) + C_{gxy}C_{fy}(\frac{1}{\eta\mu_g^2}(\frac{2\sqrt{d_2}}{\delta} + \eta\frac{2\sqrt{d_2}}{\delta}L_g)(1-\eta\mu_g) + \frac{L_{gyy}}{\mu_g^2}) + \frac{C_{gxy}}{\mu_g}L_f$ and $L_2(\frac{\sqrt{d_2}}{\delta}) = L_f + \frac{L_{gxy}C_{fy}}{\mu_g} + \frac{C_{gxy}C_{fy}}{\mu_g}\eta\frac{2\sqrt{d_2}}{\delta}L_g + C_{gxy}C_{fy}(\eta\frac{1}{\mu_g^2}\frac{2\sqrt{d_2}}{\delta}L_g(1-\eta\mu_g) + \frac{L_{gyy}}{\mu_g^2}) + \frac{C_{gxy}}{\mu_g}L_f.$*

*Proof.* We have

$$\|\nabla F_\delta(x_1) - \nabla F_\delta(x_2)\|$$

$$= \|\nabla f_\delta(x_1, y^*(x_1)) - \nabla f_\delta(x_2, y^*(x_2))\|$$

$$\leq \|\nabla f_\delta(x_1, y^*(x_1)) - \nabla f_\delta(x_1, y^*(x_2))\| + \|f_\delta(x_1, y^*(x_2)) - f_\delta(x_2, y^*(x_2))\| \tag{54}$$

For the first term, we have

$$\|\nabla f_\delta(x_1, y^*(x_1)) - \nabla f_\delta(x_1, y^*(x_2))\| \leq L_1(\frac{\sqrt{d_2}}{\delta})\|y^*(x_1) - y^*(x_2)\| \leq \frac{L_g}{\mu_g}L_1(\frac{\sqrt{d_2}}{\delta})\|x_1 - x_2\| \tag{55}$$

Thus, we have

$$\|\nabla F_\delta(x_1) - \nabla F_\delta(x_2)\| \leq \left(L_2(\frac{\sqrt{d_2}}{\delta}) + \frac{L_g}{\mu_g}L_1(\frac{\sqrt{d_2}}{\delta})\right)\|x_1 - x_2\| \tag{56}$$

$$\square$$

## G Deriving the convergence metric

In this section, we give a detailed analysis to derive the metric. Based on the Lipshitz continuous of $\nabla F_\delta(x)$, we have the following lemma,

**Lemma 9.** *Under Assumptions 1, 2 and Lemma 3, we have*

$$\|\nabla F_\delta(x_k) - w_k\|^2 \leq 2L_1^2(\frac{\sqrt{d_2}}{\delta})\|y^*(x_k) - y_k\|^2 + 2\|\nabla f_\delta(x_k, y_k) - w_k\|^2. \tag{57}$$

*Proof.* We have

$$\|\nabla F_\delta(x_k) - w_k\|^2$$

$$= \|\nabla f_\delta(x_k, y^*(x_k)) - \nabla f_\delta(x_k, y_k) + \nabla f_\delta(x_k, y_k) - w_k\|^2$$

$$\leq 2\|\nabla f_\delta(x_k, y^*(x_k)) - \nabla f_\delta(x_k, y_k)\|^2 + 2\|\nabla f_\delta(x_k, y_k) - w_k\|^2$$

$$\leq 2L_1^2(\frac{\sqrt{d_2}}{\delta})\|y^*(x_k) - y_k\|^2 + 2\|\nabla f_\delta(x_k, y_k) - w_k\|^2 \tag{58}$$

$$\square$$

Then, we give the proof of Lemma 5

*Proof.* We have

$$\|\mathcal{G}\left(x_k, \nabla F(x_k), \hat{\gamma}\right)\|$$

$$=\frac{1}{\hat{\gamma}}\|x_k - \mathcal{P}_{\mathcal{X}}\left(x_k - \hat{\gamma}\nabla F_\delta(x_k)\right) + \mathcal{P}_{\mathcal{X}}\left(x_k - \hat{\gamma}\nabla F_\delta(x_k)\right) - \mathcal{P}_{\mathcal{X}}\left(x_k - \hat{\gamma}\nabla F(x_k)\right)\|$$

$$=\frac{1}{\hat{\gamma}}\|x_k - \mathcal{P}_{\mathcal{X}}\left(x_k - \hat{\gamma}\nabla F_\delta(x_k)\right)\| + \frac{1}{\hat{\gamma}}\|\mathcal{P}_{\mathcal{X}}\left(x_k - \hat{\gamma}\nabla F_\delta(x_k)\right) - \mathcal{P}_{\mathcal{X}}\left(x_k - \hat{\gamma}\nabla F(x_k)\right)\|$$

$$\leq\frac{1}{\hat{\gamma}}\|x_k - \mathcal{P}_{\mathcal{X}}\left(x_k - \hat{\gamma}\nabla F_\delta(x_k)\right)\| + \|\nabla F_\delta(x_k) - \nabla F(x_k)\|$$

$$(59)$$

where the last inequality is due to the non-expansive of the projection operator.

Using Proposition 1 of Ghadimi et al. (2016), we can obtain

$$\|\nabla F_\delta(x_k) - \nabla F(x_k)\|$$

$$=\|\eta\nabla_{xy}^2 g(x, y^*(x))\nabla\mathcal{P}_{\mathcal{Y}}(z^*)^\top \left[I_{d_2} - (I_{d_2} - \eta\nabla_{yy}^2 g(x, y^*(x)))\nabla\mathcal{P}_{\mathcal{Y}}(z^*)^\top\right]^{-1}\nabla_y f(x, y^*(x))$$

$$- \eta\nabla_{xy}^2 g(x, y^*(x))\nabla\mathcal{P}_{\mathcal{Y}\delta}(z^*)^\top \left[I_{d_2} - (I_{d_2} - \eta\nabla_{yy}^2 g(x, y^*(x)))\nabla\mathcal{P}_{\mathcal{Y}\delta}(z^*)^\top\right]^{-1}\nabla_y f(x, y^*(x))\|$$

$$\leq\eta\|\nabla_{xy}^2 g(x, y^*(x))\|\|\nabla\mathcal{P}_{\mathcal{Y}}(z^*) - \nabla\mathcal{P}_{\mathcal{Y}\delta}(z^*)\|\|\left[I_{d_2} - (I_{d_2} - \eta\nabla_{yy}^2 g(x, y^*(x)))\nabla\mathcal{P}_{\mathcal{Y}}(z^*)^\top\right]^{-1}\|\|\nabla_y f(x, y^*(x))\|$$

$$+ \eta\|\nabla_{xy}^2 g(x, y^*(x))\|\|\nabla\mathcal{P}_{\mathcal{Y}\delta}(z^*)\|\|\left[I_{d_2} - (I_{d_2} - \eta\nabla_{yy}^2 g(x, y^*(x)))\nabla\mathcal{P}_{\mathcal{Y}}(z^*)^\top\right]^{-1}$$

$$- \left[I_{d_2} - (I_{d_2} - \eta\nabla_{yy}^2 g(x, y^*(x)))\nabla\mathcal{P}_{\mathcal{Y}\delta}(z^*)^\top\right]^{-1}\|\|\nabla_y f(x, y^*(x))\|$$

$$\leq\eta C_{gxy}\frac{\delta L}{2}(d_2 + 3)^{3/2}C_{fy}\|\left[I_{d_2} - (I_{d_2} - \eta\nabla_{yy}^2 g(x, y^*(x)))\nabla\mathcal{P}_{\mathcal{Y}}(z^*)^\top\right]^{-1}\|$$

$$+ \eta C_{gxy}C_{fy}\|\left[I_{d_2} - (I_{d_2} - \eta\nabla_{yy}^2 g(x, y^*(x)))\nabla\mathcal{P}_{\mathcal{Y}}(z^*)^\top\right]^{-1}$$

$$- \left[I_{d_2} - (I_{d_2} - \eta\nabla_{yy}^2 g(x, y^*(x)))\nabla\mathcal{P}_{\mathcal{Y}\delta}(z^*)^\top\right]^{-1}\|$$

$$(60)$$

Since $\|(I_{d_2} - \eta\nabla_{yy}^2 g(x, y^*(x)))\nabla\mathcal{P}_{\mathcal{Y}}(z^*)^\top\| \leq \|\nabla\mathcal{P}_{\mathcal{Y}}(z^*)\|\|I_{d_2} - \eta\nabla_{yy}^2 g(x, y^*(x))\| \leq 1 - \eta\mu_g \leq 1$ we have

$$\|\left[(I_{d_2} - \eta\nabla_{yy}^2 g(x, y^*(x)))\nabla\mathcal{P}_{\mathcal{Y}}(z^*)^\top\right]^{-1}\|$$

$$\leq\frac{1}{1 - \|(I_{d_2} - \eta\nabla_{yy}^2 g(x, y^*(x)))\nabla\mathcal{P}_{\mathcal{Y}}(z^*)^\top\|}$$

$$\leq\frac{1}{\eta\mu_g}$$

$$(61)$$

Using the inequality $\|H_2^{-1} - H_1^{-1}\| \leq \|H_1^{-1}(H_1 - H_2)H_2^{-1}\| \leq \|H_1^{-1}\|\|H_1 - H_2\|\|H_2^{-1}\|$, we have

$$\|\left[I_{d_2} - (I_{d_2} - \eta\nabla_{yy}^2 g(x, y^*(x)))\nabla\mathcal{P}_{\mathcal{Y}}(z^*)^\top\right]^{-1} - \left[I_{d_2} - (I_{d_2} - \eta\nabla_{yy}^2 g(x, y^*(x)))\nabla\mathcal{P}_{\mathcal{Y}\delta}(z^*)^\top\right]^{-1}\|$$

$$\leq\frac{1}{\eta^2\mu_g^2}\|(I_{d_2} - \eta\nabla_{yy}^2 g(x, y^*(x)))\nabla\mathcal{P}_{\mathcal{Y}}(z^*)^\top - (I_{d_2} - \eta\nabla_{yy}^2 g(x, y^*(x)))\nabla\mathcal{P}_{\mathcal{Y}\delta}(z^*)^\top\|$$

$$\leq\frac{1}{\eta^2\mu_g^2}\|\nabla\mathcal{P}_{\mathcal{Y}\delta}(z^*) - \nabla\mathcal{P}_{\mathcal{Y}}(z^*)\|\|I_{d_2} - \eta\nabla_{yy}^2 g(x, y^*(x))\|$$

$$\leq\frac{\delta L}{2\eta^2\mu_g^2}(d_2 + 3)^{3/2}(1 - \eta\mu_g)$$

$$(62)$$

Using the above inequalities, we have

$$\|\mathcal{G}\left(x_k, \nabla F(x_k), \hat{\gamma}\right)\|$$

$$\leq\frac{1}{\hat{\gamma}}\|x_k - \mathcal{P}_{\mathcal{X}}\left(x_k - \hat{\gamma}\nabla F_\delta(x_k)\right)\| + \frac{\delta L}{2\mu_g}C_{gxy}(d_2 + 3)^{3/2}C_{fy}(1 + \frac{1}{\mu_g}(1 - \eta\mu_g))$$

$$(63)$$

Let $\mathcal{G}\left(x_k, \nabla F_\delta(x_k), \hat{\gamma}\right) = \frac{1}{\hat{\gamma}}\left(x_k - \mathcal{P}_{\mathcal{X}}\left(x_k - \hat{\gamma}\nabla F_\delta(x_k)\right)\right)$

$$
\begin{aligned}
&\|\mathcal{G}\left(x_k, \nabla F_\delta(x_k), \hat{\gamma}\right)\| \\
\leq &\|\mathcal{G}\left(x_k, \nabla F_\delta(x_k), \hat{\gamma}\right) - \mathcal{G}(x_t, w_k, \hat{\gamma})\| + \|\mathcal{G}(x_t, w_k, \hat{\gamma})\| \\
\leq &\|\nabla F_\delta(x_k) - w_k\| + \|\mathcal{G}(x_t, w_k, \hat{\gamma})\| \\
\leq &\|w_k - \nabla f(x_k, y_k) - R_k\| + \|R_k\|^2 + L_0\|y^*(x_k) - y_k\| + \frac{1}{\gamma c_l}\|x_k - \tilde{x}_{k+1}\|.
\end{aligned}
\tag{64}
$$

where $L_0 = \max(L_1(\frac{\sqrt{d_2}}{\delta}), L_2(\frac{\sqrt{d_2}}{\delta}))$ and $R_k = \nabla f_\delta(x, y) - \mathbb{E}[\bar{\nabla} f_\delta(x, y; \bar{\xi})]$.

Therefore, we have

$$
\begin{aligned}
\mathcal{G}\left(x_k, \nabla F(x_k), \hat{\gamma}\right) \leq &\|w_k - \nabla f(x_k, y_k) - R_k\| + \|R_k\| + L_0\|y^*(x_k) - y_k\| + \frac{1}{\gamma c_l}\|x_k - \tilde{x}_{k+1}\| \\
&+ \frac{\delta L}{2\mu_g}C_{gxy}(d_2 + 3)^{3/2}C_{fy}(1 + \frac{1}{\mu_g}(1 - \eta\mu_g)).
\end{aligned}
\tag{65}
$$

$\square$

## G.1 USEFULL LEMMAS IN CONVERGENCE RATE

**Lemma 10.** (***Descent on the function value.***) *Under Assumptions 1, 2, and Lemma 3, let $F_\delta(x)$ be an approximation function of $F(x)$ and have the gradient $\nabla F_\delta(x_k)$, and $\gamma\eta_k \leq \dfrac{1}{2L_{F_\delta}(\frac{d_2}{\delta})c_l}$, we have*

$$
F_\delta(x_{k+1}) \leq F_\delta(x_k) + \eta_k\gamma c_l\|\nabla F_\delta(x_k) - w_k\|^2 - \frac{\eta_k}{2\gamma c_l}\|\tilde{x}_{k+1} - x_k\|^2
\tag{66}
$$

*Proof.* Due to the smoothness of $F_\delta$ and $\tilde{x}_{k+1} = \mathcal{P}_{\mathcal{X}}(x_k - \frac{\gamma}{\sqrt{\|w_k\|} + G_0}w_k)$, we have

$$
\begin{aligned}
&F_\delta(x_{k+1}) \\
\leq &F_\delta(x_k) + \nabla F_\delta(x_k)^\top(x_{k+1} - x_k) + \frac{1}{2}L_{F_\delta}(\frac{\sqrt{d_2}}{\delta})\|x_{k+1} - x_k\|^2 \\
= &F_\delta(x_k) + \eta_k\nabla F_\delta(x_k)^\top(\tilde{x}_{k+1} - x_k) + \frac{1}{2}L_{F_\delta}(\frac{\sqrt{d_2}}{\delta})\|\eta_k(\tilde{x}_{k+1} - x_k)\|^2 \\
= &F_\delta(x_k) + \eta_k\langle w_k, \tilde{x}_{k+1} - x_k\rangle + \eta_k\langle\nabla F_\delta(x_k) - w_k, \tilde{x}_{k+1} - x_k\rangle + \frac{\eta_k^2}{2}L_{F_\delta}(\frac{\sqrt{d_2}}{\delta})\|\tilde{x}_{k+1} - x_k\|^2
\end{aligned}
\tag{67}
$$

In our algorithm, we have $\tilde{x}_{k+1} = \mathcal{P}_{\mathcal{X}}(x_k - \frac{\gamma}{\sqrt{\|w_k\|} + G_0}w_k) = \arg\min_{x\in\mathcal{X}}\frac{1}{2}\|x - x_k + \frac{\gamma}{\sqrt{\|w_k\|} + G_0}w_k\|^2$. We have the following optimal condition,

$$
\langle\tilde{x}_{k+1} - x_k + \frac{\gamma}{\sqrt{\|w_k\|} + G_0}w_k, x - \tilde{x}_{k+1}\rangle \geq 0, \; x \in \mathcal{X}
\tag{68}
$$

Set $x = x_k$, we can obtain

$$
\gamma c_l\langle w_k, \tilde{x}_{k+1} - x_k\rangle \leq \frac{\gamma}{\sqrt{\|w_k\|} + G_0}\langle w_k, \tilde{x}_{k+1} - x_k\rangle \leq -\|\tilde{x}_{k+1} - x_k\|^2
\tag{69}
$$

Thus, we have

$$
\langle w_k, \tilde{x}_{k+1} - x_k\rangle \leq -\frac{1}{\gamma c_l}\|\tilde{x}_{k+1} - x_k\|^2
\tag{70}
$$

In addition, we can obtain

$$\langle \nabla F_\delta(x_k) - w_k, \tilde{x}_{k+1} - x_k \rangle$$
$$\leq \|\nabla F_\delta(x_k) - w_k\|_2 \|\tilde{x}_{k+1} - x_k\|_2$$
$$\leq \gamma c_l \|\nabla F_\delta(x_k) - w_k\|^2 + \frac{1}{4\gamma c_l} \|\tilde{x}_{k+1} - x_k\|^2 \tag{71}$$

Then, setting $\gamma \leq \dfrac{1}{2L_{F_\delta}(\frac{d_2}{\delta})c_l\eta_k}$, we can derive

$$F_\delta(x_{k+1})$$

$$\leq F_\delta(x_k) + \eta_k\gamma c_l\|\nabla F_\delta(x_k) - w_k\|^2 + \frac{\eta_k}{4\gamma c_l}\|\tilde{x}_{k+1} - x_k\|^2 - \frac{\eta_k}{\gamma c_l}\|\tilde{x}_{k+1} - x_k\|^2 + \frac{\eta_k^2}{2}L_{F_\delta}(\frac{\sqrt{d_2}}{\delta})\|\tilde{x}_{k+1} - x_k\|^2$$

$$\leq F_\delta(x_k) + \eta_k\gamma c_l\|\nabla F_\delta(x_k) - w_k\|^2 - \frac{\eta_k}{2\gamma c_l}\|\tilde{x}_{k+1} - x_k\|^2 \tag{72}$$

$\square$

**Lemma 11.** *(**Error between the updates of** $y$ **and the optimal solution** $y^*$) Under Assumptions 1, 2, 4, and Lemma 3, let* $y_{k+1} = (1 - \eta_k)y_k + \eta_k\mathcal{P}_{\mathcal{Y}}(y_k - \dfrac{\tau}{\sqrt{\|v_k\|} + G_0}v_k)$, $\eta_k \leq 1$, $\tau \leq \dfrac{1}{6L_gc_u}$, *we have*

$$\|y_{k+1} - y^*(x_{k+1})\|^2$$
$$\leq (1 - \frac{\eta_k\tau\mu_g c_u}{4})\|y^*(x_k) - y_k\|^2 + \frac{25\eta_k\tau c_u}{6\mu_g}\|\nabla_y g(x_k, y_k) - v_k\|^2$$
$$- \frac{3\eta_k}{4}\|\tilde{y}_{k+1} - y_k\|^2 + \frac{25L_y^2\eta_k}{6\tau\mu_g c_u}\|x_k - \tilde{x}_{k+1}\|^2 \tag{73}$$

*Proof.* Define $\tilde{y}_{k+1} = \mathcal{P}_{\mathcal{Y}}(y_k - \dfrac{\tau}{\sqrt{\|v_k\|} + G_0}v_k)$. We have $y_{k+1} = (1 - \eta_k)y_k + \eta_k\tilde{y}_{k+1}$.

According to the strong convexity of $g$, we have

$$g(x_k, y)$$
$$\geq g(x_k, y_k) + \langle \nabla_y g(x_k, y_k), y - y_k \rangle + \frac{\mu_g}{2}\|y - y_k\|^2$$
$$= g(x_k, y_k) + \langle v_k, y - \tilde{y}_{k+1} \rangle + \langle \nabla_y g(x_k, y_k) - v_k, y - \tilde{y}_{k+1} \rangle + \langle \nabla_y g(x_k, y_k), \tilde{y}_{k+1} - y_k \rangle + \frac{\mu_g}{2}\|y - y_k\|^2 \tag{74}$$

According to the smoothness of $g$, we have

$$g(x_k, \tilde{y}_{k+1}) \leq g(x_k, y_k) + \langle \nabla_y g(x_k, y_k), \tilde{y}_{k+1} - y_k \rangle + \frac{L_g}{2}\|\tilde{y}_{k+1} - y_k\|^2 \tag{75}$$

Then, combining the above inequalities, we have

$$g(x_k, y) \geq g(x_k, \tilde{y}_{k+1}) - \frac{L_g}{2}\|\tilde{y}_{k+1} - y_k\|^2 + \langle v_k, y - \tilde{y}_{k+1} \rangle + \langle \nabla_y g(x_k, y_k) - v_k, y - \tilde{y}_{k+1} \rangle + \frac{\mu_g}{2}\|y - y_k\|^2 \tag{76}$$

In our Algorithm 1, we have

$$\tilde{y}_{k+1} = \mathcal{P}_{\mathcal{Y}}\left(y_k - \frac{\tau}{\sqrt{\|v_k\|} + G_0}v_k\right) = \arg\min_{y\in\mathcal{Y}}\frac{1}{2}\left\|y - y_k + \frac{\tau}{\sqrt{\|v_k\|} + G_0}v_k\right\|^2. \tag{77}$$

Since $\mathcal{Y}$ is a convex set and the function $\dfrac{1}{2}\left\|y - y_k + \dfrac{\tau}{\sqrt{\|v_k\|} + G_0}v_k\right\|^2$ is convex, we have

$$\left\langle \tilde{y}_{k+1} - y_k + \frac{\tau}{\sqrt{\|v_k\|} + G_0}v_k, y - \tilde{y}_{k+1} \right\rangle \geq 0, \quad y \in \mathcal{Y} \tag{78}$$

Then we have

$$\tau c_u \langle v_k, y - \tilde{y}_{k+1} \rangle \geq \frac{\tau}{\sqrt{\|v_k\|} + G_0} \langle v_k, y - \tilde{y}_{k+1} \rangle \geq \langle \tilde{y}_{k+1} - y_k, \tilde{y}_{k+1} - y \rangle \tag{79}$$

Then we have

$$\begin{aligned}
&g(x_k, y) \\
&\geq g(x_k, \tilde{y}_{k+1}) - \frac{L_g}{2} \|\tilde{y}_{k+1} - y_k\|^2 + \frac{\mu_g}{2} \|y - y_k\|^2 + \langle \nabla_y g(x_k, y_k) - v_k, y - \tilde{y}_{k+1} \rangle \\
&\quad + \frac{1}{\tau c_u} \langle \tilde{y}_{k+1} - y_k, \tilde{y}_{k+1} - y \rangle
\end{aligned} \tag{80}$$

Let $y = y^*(x_k)$. Since $g(x_k, y^*(x_k)) \leq g(x_k, \tilde{y}_{k+1})$, we have

$$\begin{aligned}
&g(x_k, \tilde{y}_{k+1}) \geq g(x_k, y^*(x_k)) \\
&\geq g(x_k, \tilde{y}_{k+1}) - \frac{L_g}{2} \|\tilde{y}_{k+1} - y_k\|^2 + \frac{\mu_g}{2} \|y^*(x_k) - y_k\|^2 + \langle \nabla_y g(x_k, y_k) - v_k, y^*(x_k) - \tilde{y}_{k+1} \rangle \\
&\quad + \frac{1}{\tau c_u} \|\tilde{y}_{k+1} - y_k\|^2 + \frac{1}{\tau c_u} \langle \tilde{y}_{k+1} - y_k, y_k - y^*(x_k) \rangle
\end{aligned} \tag{81}$$

In addition, we have

$$\begin{aligned}
&\langle \nabla_y g(x_k, y_k) - v_k, y^*(x_k) - \tilde{y}_{k+1} \rangle \\
&= \langle \nabla_y g(x_k, y_k) - v_k, y^*(x_k) - y_k \rangle + \langle \nabla_y g(x_k, y_k) - v_k, y_k - \tilde{y}_{k+1} \rangle \\
&\geq -\|\nabla_y g(x_k, y_k) - v_k\| \|y^*(x_k) - y_k\| - \|\nabla_y g(x_k, y_k) - v_k\| \|y_k - \tilde{y}_{k+1}\| \\
&\geq -\frac{1}{\mu_g} \|\nabla_y g(x_k, y_k) - v_k\|^2 - \frac{\mu_g}{4} \|y^*(x_k) - y_k\|^2 - \frac{1}{\mu_g} \|\nabla_y g(x_k, y_k) - v_k\|^2 - \frac{\mu_g}{4} \|y_k - \tilde{y}_{k+1}\|^2 \\
&\geq -\frac{2}{\mu_g} \|\nabla_y g(x_k, y_k) - v_k\|^2 - \frac{\mu_g}{4} \|y^*(x_k) - y_k\|^2 - \frac{\mu_g}{4} \|y_k - \tilde{y}_{k+1}\|^2
\end{aligned} \tag{82}$$

The first inequality is due to $\langle a, b \rangle \geq -\|a\| \|b\|$ and the second inequality is due to the Young's inequality. We also have

$$\begin{aligned}
&\|y_{k+1} - y^*(x_k)\|^2 \\
&\leq \|y_k + \eta_k(\tilde{y}_{k+1} - y_k) - y^*(x_k)\|^2 \\
&= \|y_k - y^*(x_k)\|^2 + \eta_k^2 \|\tilde{y}_{k+1} - y_k\|^2 + 2\eta_k \langle \tilde{y}_{k+1} - y_k, y_k - y^*(x_k) \rangle
\end{aligned} \tag{83}$$

Therefore, we have

$$\langle \tilde{y}_{k+1} - y_k, y_k - y^*(x_k) \rangle \geq \frac{1}{2\eta_k}(\|y_{k+1} - y^*(x_k)\|^2 - \|y_k - y^*(x_k)\|^2 - \eta_k^2 \|\tilde{y}_{k+1} - y_k\|^2) \tag{84}$$

Then, we have

$$\begin{aligned}
0 \geq &-\frac{L_g}{2} \|\tilde{y}_{k+1} - y_k\|^2 + \frac{\mu_g}{2} \|y^*(x_k) - y_k\|^2 + \frac{1}{\tau c_u} \|\tilde{y}_{k+1} - y_k\|^2 \\
&- \frac{2}{\mu_g} \|\nabla_y g(x_k, y_k) - v_k\|^2 - \frac{\mu_g}{4} \|y^*(x_k) - y_k\|^2 - \frac{\mu_g}{4} \|y_k - \tilde{y}_{k+1}\|^2 \\
&+ \frac{1}{2\eta_k \tau c_u}(\|y_{k+1} - y^*(x_k)\|^2 - \|y_k - y^*(x_k)\|^2 - \eta_k^2 \|\tilde{y}_{k+1} - y_k\|^2)
\end{aligned} \tag{85}$$

Hence we have

$$
\begin{aligned}
\|y_{k+1} - y^*(x_k)\|^2 \leq &2\eta_k \tau c_u (\frac{L_g}{2} - \frac{1}{\tau c_u} + \frac{\mu_g}{4} + \frac{\eta_k}{2\tau c_u})\|\tilde{y}_{k+1} - y_k\|^2 \\
&+ (1 - \frac{\mu_g \eta_k \tau c_u}{2})\|y^*(x_k) - y_k\|^2 + \frac{4\eta_k \tau c_u}{\mu_g}\|\nabla_y g(x_k, y_k) - v_k\|^2 \\
\leq &(1 - \frac{\mu_g \eta_k \tau c_u}{2})\|y^*(x_k) - y_k\|^2 + \frac{4\eta_k \tau c_u}{\mu_g}\|\nabla_y g(x_k, y_k) - v_k\|^2 \\
&- 2\eta_k \tau c_u (\frac{1}{2\tau c_u} - \frac{3L_g}{4})\|\tilde{y}_{k+1} - y_k\|^2 \\
\leq &(1 - \frac{\mu_g \eta_k \tau c_u}{2})\|y^*(x_k) - y_k\|^2 + \frac{4\eta_k \tau c_u}{\mu_g}\|\nabla_y g(x_k, y_k) - v_k\|^2 \\
&- \frac{3\eta_k}{4}\|\tilde{y}_{k+1} - y_k\|^2
\end{aligned}
\tag{86}
$$

using $\eta_k \leq 1$, $\mu_g \leq L_g$ and $\tau \leq \frac{1}{6L_g c_u}$.

Then, we have

$$
\begin{aligned}
&\|y_{k+1} - y^*(x_{k+1})\|^2 \\
=&\|y_{k+1} - y^*(x_k) + y^*(x_k) - y^*(x_{k+1})\|^2 \\
\leq&(1 + \frac{\eta_k \tau \mu_g c_u}{4})\|y_{k+1} - y^*(x_k)\|^2 + (1 + \frac{4}{\eta_k \tau \mu_g c_u})\|y^*(x_k) - y^*(x_{k+1})\|^2 \\
\leq&(1 + \frac{\eta_k \tau \mu_g c_u}{4})\|y_{k+1} - y^*(x_k)\|^2 + (1 + \frac{4}{\eta_k \tau \mu_g c_u})L_y^2\|x_k - x_{k+1}\|^2 \\
\leq&(1 + \frac{\eta_k \tau \mu_g c_u}{4})(1 - \frac{\mu_g \eta_k \tau c_u}{2})\|y^*(x_k) - y_k\|^2 + (1 + \frac{\eta_k \tau \mu_g c_u}{4})\frac{4\eta_k \tau c_u}{\mu_g}\|\nabla_y g(x_k, y_k) - v_k\|^2 \\
&- (1 + \frac{\eta_k \tau \mu_g c_u}{4})\frac{3\eta_k}{4}\|\tilde{y}_{k+1} - y_k\|^2 + (1 + \frac{4}{\eta_k \tau \mu_g c_u})L_y^2\|x_k - x_{k+1}\|^2
\end{aligned}
\tag{87}
$$

Since $\eta_k \leq 1$, $\mu_g \leq L_g$ and $\tau \leq \frac{1}{6L_g c_u}$, we have $\tau \leq \frac{1}{6L_g c_u} \leq \frac{1}{6\mu_g c_u}$ and $\eta_k \leq 1 \leq \frac{1}{6\tau L_g c_u}$.
Then, we can obtain

$$
(1 + \frac{\eta_k \tau \mu_g c_u}{4})(1 - \frac{\mu_g \eta_k \tau c_u}{2}) = 1 - \frac{\mu_g \eta_k \tau c_u}{2} + \frac{\eta_k \tau \mu_g c_u}{4} - \frac{\eta_k^2 \tau^2 \mu_g^2 c_u^2}{8} \leq 1 - \frac{\eta_k \tau \mu_g c_u}{4}
\tag{88}
$$

$$
-(1 + \frac{\eta_k \tau \mu_g c_u}{4})\frac{3\eta_k}{4} \leq -\frac{3\eta_k}{4}
\tag{89}
$$

$$
(1 + \frac{\eta_k \tau \mu_g c_u}{4})\frac{4\eta_k \tau c_u}{\mu_g} \leq \frac{25\eta_k \tau c_u}{6\mu_g}
\tag{90}
$$

$$
(1 + \frac{4}{\eta_k \tau \mu_g c_u})L_y^2 \leq \frac{25L_y^2}{6\eta_k \tau \mu_g c_u}
\tag{91}
$$

Finally, we can obtain

$$
\begin{aligned}
&\|y_{k+1} - y^*(x_{k+1})\|^2 \\
\leq&(1 - \frac{\eta_k \tau \mu_g c_u}{4})\|y^*(x_k) - y_k\|^2 + \frac{25\eta_k \tau c_u}{6\mu_g}\|\nabla_y g(x_k, y_k) - v_k\|^2 \\
&- \frac{3\eta_k}{4}\|\tilde{y}_{k+1} - y_k\|^2 + \frac{25L_y^2}{6\eta_k \tau \mu_g c_u}\|x_k - x_{k+1}\|^2 \\
\leq&(1 - \frac{\eta_k \tau \mu_g c_u}{4})\|y^*(x_k) - y_k\|^2 + \frac{25\eta_k \tau c_u}{6\mu_g}\|\nabla_y g(x_k, y_k) - v_k\|^2 \\
&- \frac{3\eta_k}{4}\|\tilde{y}_{k+1} - y_k\|^2 + \frac{25L_y^2 \eta_k}{6\tau \mu_g c_u}\|x_k - \tilde{x}_{k+1}\|^2
\end{aligned}
\tag{92}
$$

$\square$

**Lemma 12.** *(Descent in the gradient estimation error.(Huang & Huang, 2021)) Under Assumptions 1, 2, 4, and Lemma 4, if $\alpha \in (0,1)$ and $\beta \in (0,1)$, we have*

$$\mathbb{E}[\|\nabla f_\delta(x_{k+1}, y_{k+1}) + R_{k+1} - w_{k+1}\|^2]$$

$$\leq (1-\alpha)\mathbb{E}[\|\nabla f_\delta(x_k, y_k) + R_k - w_k\|^2] + \alpha^2 \sigma_f^2(d_2) + \frac{3}{\alpha}(\|R_k\|^2 + \|R_{k+1}\|^2)$$

$$+ \frac{3}{\alpha}L_0^2\eta_k^2(\|x_k - \tilde{x}_{k+1}\|^2 + \|y_k - \tilde{y}_{k+1}\|^2) \tag{93}$$

$$\mathbb{E}[\|\nabla g(x_{k+1}, y_{k+1}) - v_{k+1}\|^2]$$

$$\leq (1-\beta)\mathbb{E}[\|\nabla g(x_k, y_k) - v_k\|^2] + \frac{2L_g^2}{\beta}\eta_k^2(\|x_k - \tilde{x}_{k+1}\|^2 + \|y_k - \tilde{y}_{k+1}\|^2) \tag{94}$$

*where $L_0 = \max(L_1(\frac{\sqrt{d_2}}{\delta}), L_2(\frac{\sqrt{d_2}}{\delta}))$.*

## G.2 PROOF OF THE CONVERGENCE RATE IN THEOREM 1

Here we first give a detailed version of Theorem 1 and then present the proof.

**Theorem 2.** *Under Assumptions 1, 2 4 and Lemma 3, with $\frac{1}{\mu_g}(1 - \frac{1}{d_2+4}) \leq \eta \leq \frac{1}{\mu_g}$, $Q = \frac{1}{\mu_g\eta}\ln\frac{C_{gxy}C_{fy}K}{\mu_g}$, $\gamma \leq \min\{\frac{1}{L_0^{2-a}}, \frac{1}{4c_l\left(\frac{125L_0^aL_y^2c_l\eta_k}{3\tau^2\mu_g^2c_u^2} + (\frac{2}{3}L_0^2 + \frac{6\mu_g^2L_g^2}{125L_0^2})c_l\right)}, 1\}$ $0 < \tau < \frac{15L_0^a}{2\mu_gc_u\left(\frac{2}{3}L_0^2 + \frac{6\mu_g^2L_g^2}{125L_0^2}\right)}$,, $m \geq \max\{t^2, c_1^2t^2, c_2^2t^2\}$, $\alpha = c_1\eta_k$, $\beta = c_2\eta_k$, $\frac{9}{2} \leq c_1 \leq \frac{m^{1/2}}{t}$, $\frac{125L_0^2}{3\mu_g^2} \leq c_2 \leq \frac{m^{1/2}}{t}$, $L_0 = \max(L_1(\frac{\sqrt{d_2}}{\delta}) > 1, L_2(\frac{\sqrt{d_2}}{\delta}))$, $\Phi_1 = F_\delta(x_1) + \frac{10L_0^ac_l}{\tau\mu_gc_u}\|y_1 - y^*(x_1)\|^2 + c_l(\|w_1 - \bar{\nabla}f_\delta(x_1, y_{k+1}) - R_1\|^2 + \|\nabla_y g(x_1, y_1) - v_1\|^2)$, $G = \frac{\Phi_1 - \Phi^*}{\gamma c_l} + \frac{17t}{4K^2}(m + K)^{1/2} + \frac{4}{3tK^2}(m + K)^{3/2} + (m\sigma_f^2(d_2))t^2\ln(m + K)$, and $\eta_k = \frac{t}{(m+k)^{1/2}}$, $t > 0$, we have*

$$\frac{1}{K}\sum_{k=1}^{K}\mathbb{E}[\frac{1}{2}\mathcal{M}_k] \leq \frac{2m^{1/4}\sqrt{G}}{\sqrt{Kt}} + \frac{2\sqrt{G}}{(Kt)^{1/4}} + \frac{\delta L}{4\mu_g}C_{gxy}(d_2 + 3)^{3/2}C_{fy}(1 + \frac{1}{\mu_g}(1 - \eta\mu_g)) \tag{95}$$

*Proof.* Setting $\eta_k = \frac{t}{(m+k)^{1/2}}$ and $m \geq t^2$, we have $\eta_k \leq 1$. Due to $m \geq (c_1t)^2$, we have $\alpha = c_1\eta_k \leq \frac{c_1t}{m^{1/2}} \leq 1$. Due to $m \geq (c_2t)^2$, we have $\beta = c_2\eta_k \leq \frac{c_2t}{m^{1/2}} \leq 1$. Also, we have $c_1, c_2 \leq \frac{m^{1/2}}{t}$. Then using the above lemmas, we have

$$\mathbb{E}[\|\nabla f_\delta(x_{k+1}, y_{k+1}) + R_{k+1} - w_{k+1}\|^2] - \mathbb{E}[\|\nabla f_\delta(x_k, y_k) + R_k - w_k\|^2]$$

$$\leq -\alpha\mathbb{E}[\|\nabla f_\delta(x_k, y_k) + R_k - w_k\|^2] + \alpha^2\sigma_f^2(d_2) + \frac{3}{\alpha}(\|R_k\|^2 + \|R_{k+1}\|^2)$$

$$+ \frac{3}{\alpha}L_0^2\eta_k^2(\|x_k - \tilde{x}_{k+1}\|^2 + \|y_k - \tilde{y}_{k+1}\|^2)$$

$$\leq -c_1\eta_k\mathbb{E}[\|\nabla f_\delta(x_k, y_k) + R_k - w_k\|^2] + c_1^2\eta_k^2\sigma_f^2(d_2) + \frac{3}{c_1\eta_k}(\|R_k\|^2 + \|R_{k+1}\|^2)$$

$$+ \frac{3}{c_1}L_0^2\eta_k(\|x_k - \tilde{x}_{k+1}\|^2 + \|y_k - \tilde{y}_{k+1}\|^2)$$

$$\leq -\frac{9}{2}\eta_k\mathbb{E}[\|\nabla f_\delta(x_k, y_k) + R_k - w_k\|^2] + \frac{m}{t^2}\eta_k^2\sigma_f^2(d_2) + \frac{2}{3\eta_k}(\|R_k\|^2 + \|R_{k+1}\|^2)$$

$$+ \frac{2}{3}L_0^2\eta_k(\|x_k - \tilde{x}_{k+1}\|^2 + \|y_k - \tilde{y}_{k+1}\|^2) \tag{96}$$

where the last inequality holds by $\frac{9}{2} \leq c_1 \leq \frac{m^{1/2}}{t}$.

$$\mathbb{E}[\|\nabla g(x_{k+1}, y_{k+1}) - v_{k+1}\|^2] - \mathbb{E}[\|\nabla g(x_k, y_k) - w_k\|^2]$$

$$\leq -\beta\mathbb{E}[\|\nabla g(x_k, y_k) - v_k\|^2] + \frac{2L_g^2}{\beta}\eta_k^2(\|x_k - \tilde{x}_{k+1}\|^2 + \|y_k - \tilde{y}_{k+1}\|^2)$$

$$\leq -c_2\eta_k\mathbb{E}[\|\nabla g(x_k, y_k) - v_k\|^2] + \frac{2L_g^2}{c_2}\eta_k(\|\tilde{x}_k - x_{k+1}\|^2 + \|\tilde{y}_k - y_{k+1}\|^2)$$

$$\leq -\frac{125L_0^2}{3\mu_g^2}\eta_k\mathbb{E}[\|\nabla g(x_k, y_k) - v_k\|^2] + \frac{6\mu_g^2 L_g^2}{125L_0^2}\eta_k(\|x_k - \tilde{x}_{k+1}\|^2 + \|y_k - \tilde{y}_{k+1}\|^2) \tag{97}$$

where the last inequality hold by $\frac{125L_0^2}{3\mu_g^2} \leq c_2 \leq \frac{m^{1/2}}{t}$.

In addition, we have

$$F_\delta(x_{k+1}) - F_\delta(x_k)$$

$$\leq \eta_k\gamma c_l\left(2L_1^2(\frac{\sqrt{d_2}}{\delta})\|y^*(x_k) - y_k\|^2 + 2\|\nabla f_\delta(x_k, y_k) - w_k\|^2\right) - \frac{\eta_k}{2\gamma c_l}\|\tilde{x}_{k+1} - x_k\|^2$$

$$\leq 2\eta_k\gamma c_l L_0^2\|y^*(x_k) - y_k\|^2 + 2\eta_k\gamma c_l\|\nabla f_\delta(x_k, y_k) - w_k\|^2 - \frac{\eta_k}{2\gamma c_l}\|\tilde{x}_{k+1} - x_k\|^2$$

$$\leq 2\eta_k\gamma c_l L_0^2\|y^*(x_k) - y_k\|^2 + 4\eta_k\gamma c_l\|\nabla f_\delta(x_k, y_k) - w_k - R_k\|^2 + 4\eta_k\gamma c_l\|R_k\|^2 - \frac{\eta_k}{2\gamma c_l}\|\tilde{x}_{k+1} - x_k\|^2 \tag{98}$$

We can also have

$$\|y_{k+1} - y^*(x_{k+1})\|^2 - \|y^*(x_k) - y_k\|^2$$

$$\leq -\frac{\eta_k\tau\mu_g c_u}{4}\|y^*(x_k) - y_k\|^2 + \frac{25\eta_k\tau c_u}{6\mu_g}\|\nabla_y g(x_k, y_k) - v_k\|^2$$

$$-\frac{3\eta_k}{4}\|\tilde{y}_{k+1} - y_k\|^2 + \frac{25L_y^2\eta_k}{6\tau\mu_g c_u}\|x_k - \tilde{x}_{k+1}\|^2 \tag{99}$$

Then, we define a Lyapunov function, for any $k \geq 1$,

$$\Phi_{k+1}$$

$$= \mathbb{E}[F_\delta(x_{k+1}) + \frac{10L_0^a c_l}{\tau\mu_g c_u}\|y_{k+1} - y^*(x_{k+1})\|^2 + c_l(\|w_{k+1} - \bar{\nabla}f_\delta(x_{k+1}, y_{k+1}) - R_{k+1}\|^2$$

$$+ \|\nabla_y g(x_{k+1}, y_{k+1}) - v_{k+1}\|^2)] \tag{100}$$

We have

$$\Phi_{k+1} - \Phi_k$$

$$=\mathbb{E}[F_\delta(x_{k+1}) - F_\delta(x_k)] + \frac{10L_0^a c_l}{\tau\mu c_u}\mathbb{E}[\|y_{k+1} - y^*(x_{k+1})\|^2 - \|y_k - y^*(x_k)\|^2]$$

$$\quad + c_l\mathbb{E}[\|w_{k+1} - \bar{\nabla} f(x_{k+1}, y_{k+1}) - R_{k+1}\|^2 - \|w_k - \bar{\nabla} f(x_k, y_k) - R_k\|^2]$$

$$\quad + c_l\mathbb{E}[\|\nabla_y g(x_{k+1}, y_{k+1}) - v_{k+1}\|^2 - \|\nabla_y g(x_k, y_k) - v_k\|^2]$$

$$\leq 2\eta_k\gamma c_l L_0^2\|y^*(x_k) - y_k\|^2 + 4\eta_k\gamma c_l\|\nabla f_\delta(x_k, y_k) - w_k - R_k\|^2 + 4\eta_k\gamma c_l\|R_k\|^2 - \frac{\eta_k}{2\gamma c_l}\|\tilde{x}_{k+1} - x_k\|^2$$

$$\quad + \frac{10L_0^a c_l}{\tau\mu_g c_u}\left(-\frac{\eta_k\tau\mu_g c_u}{4}\|y^*(x_k) - y_k\|^2 + \frac{25\eta_k\tau c_u}{6\mu_g}\|\nabla_y g(x_k, y_k) - v_k\|^2 - \frac{3\eta_k}{4}\|\tilde{y}_{k+1} - y_k\|^2 + \frac{25L_y^2\eta_k}{6\tau\mu_g c_u}\|x_k - \tilde{x}_{k+1}\|^2\right)$$

$$\quad + c_l\left(-\frac{9}{2}\eta_k\mathbb{E}[\|\nabla f_\delta(x_k, y_k) + R_k - w_k\|^2] + \frac{m}{t^2}\eta_k^2\sigma_f^2(d_2) + \frac{2}{3\eta_k}(\|R_k\|^2 + \|R_{k+1}\|^2)\right.$$

$$\quad + \frac{2}{3}L_0^2\eta_k(\|x_k - \tilde{x}_{k+1}\|^2 + \|y_k - \tilde{y}_{k+1}\|^2))$$

$$\quad + c_l\left(-\frac{125L_0^2}{3\mu_g^2}\eta_k\mathbb{E}[\|\nabla g(x_k, y_k) - v_k\|^2] + \frac{6\mu_g^2 L_g^2}{125L_0^2}\eta_k(\|x_k - \tilde{x}_{k+1}\|^2 + \|y_k - \tilde{y}_{k+1}\|^2)\right)$$

$$\leq(2\eta_k\gamma c_l L_0^2 - \frac{5L_0^a c_l\eta_k}{2})\|y^*(x_k) - y_k\|^2 + (4\eta_k\gamma c_l - \frac{9c_l}{2}\eta_k)\|\nabla f_\delta(x_k, y_k) - w_k - R_k\|^2$$

$$\quad + (\frac{125L_0^a c_l\eta_k}{3\mu_g^2} - c_l\frac{125L_0^2}{3\mu_g^2}\eta_k)\|\nabla_y g(x_k, y_k) - v_k\|^2$$

$$\quad + \left(-\frac{\eta_k}{2\gamma c_l} + \frac{125L_0^a L_y^2 c_l\eta_k}{3\tau^2\mu_g^2 c_u^2} + (\frac{2}{3}L_0^2 + \frac{6\mu_g^2 L_g^2}{125L_0^2})\eta_k c_l\right)\|\tilde{x}_{k+1} - x_k\|^2$$

$$\quad + \left(-\frac{15L_0^a c_l\eta_k}{2\tau\mu_g c_u} + (\frac{2}{3}L_0^2 + \frac{6\mu_g^2 L_g^2}{125L_0^2})\eta_k c_l\right)\|\tilde{y}_{k+1} - y_k\|^2$$

$$\quad + 4\eta_k\gamma c_l\|R_k\|^2 + \frac{2\gamma c_l}{3\eta_k}(\|R_k\|^2 + \|R_{k+1}\|^2) + \frac{m}{t^2}\gamma c_l\eta_k^2\sigma_f^2(d_2)$$

$$\leq -\frac{L_0^2\gamma c_l\eta_k}{2}\|y^*(x_k) - y_k\|^2 - \frac{\gamma c_l}{2}\eta_k\|\nabla f_\delta(x_k, y_k) - w_k - R_k\|^2$$

$$\quad - \frac{\eta_k}{4\gamma c_l}\|\tilde{x}_{k+1} - x_k\|^2 + 4\eta_k\gamma c_l\|R_k\|^2 + \frac{2\gamma c_l}{3\eta_k}(\|R_k\|^2 + \|R_{k+1}\|^2)$$

$$\quad + \frac{m}{t^2}\gamma c_l\eta_k^2\sigma_f^2(d_2)$$

$$(101)$$

where the last inequality is due to $\gamma \leq \min\{\frac{1}{L_0^{2-a}}, \frac{1}{4c_l\left(\frac{125L_0^a L_y^2 c_l\eta_k}{3\tau^2\mu_g^2 c_u^2} + (\frac{2}{3}L_0^2 + \frac{6\mu_g^2 L_g^2}{125L_0^2})c_l\right)}, 1\}$

$0 < \tau < \frac{15L_0^a}{2\mu_g c_u\left(\frac{2}{3}L_0^2 + \frac{6\mu_g^2 L_g^2}{125L_0^2}\right)}$ and $L_0 > 1$.

Then, rearranging the above inequality, we have

$$\frac{\gamma c_l\eta_k}{4}\left(2L_0^2\|y^*(x_k) - y_k\|^2 + 2\|\nabla f_\delta(x_k, y_k) - w_k - R_k\|^2 + \|R_k\|^2 + \frac{1}{\gamma^2 c_l^2}\|\tilde{x}_{k+1} - x_k\|^2\right)$$

$$\leq\frac{17}{4}\eta_k\gamma c_l\|R_k\|^2 + \frac{2\gamma c_l}{3\eta_k}(\|R_k\|^2 + \|R_{k+1}\|^2) + \frac{m}{t^2}\gamma c_l\eta_k^2\sigma_f^2(d_2) + \Phi_k - \Phi_{k+1} \quad (102)$$

Taking the average over $k = 1, \cdots, K$ on both sides and using $\eta_k \geq \eta_K$, $Q = \dfrac{1}{\mu_g \eta} \ln \dfrac{C_{gxy} C_{fy} K}{\mu_g}$, $\eta_k = \dfrac{t}{(m+k)^{1/2}}$ and $\Phi_1 = \mathbb{E}[F_\delta(x_1) + \dfrac{10 L_0^a c_l}{\tau \mu_g c_u} \|y_1 - y^*(x_1)\|^2 + c_l(\|w_1 - \bar{\nabla} f_\delta(x_1, y_{k+1}) - R_1\|^2 + \|\nabla_y g(x_1, y_1) - v_1\|^2)]$, we have

$$\frac{1}{K} \sum_{k=1}^{K} \mathbb{E}[\frac{1}{4}\left(2 L_0^2 \|y^*(x_k) - y_k\|^2 + 2\|\nabla f_\delta(x_k, y_k) - w_k - R_k\|^2 + \|R_k\|^2 + \frac{1}{\gamma^2 c_l^2}\|\tilde{x}_{k+1} - x_k\|^2\right)]$$

$$\leq \frac{1}{K \eta_K}\left(\frac{\Phi_1 - \Phi^*}{\gamma c_l} + \frac{17}{4K^2}\sum_{k=1}^{K}\eta_k + \frac{4}{3K^2}\sum_{k=1}^{K}\frac{1}{\eta_k} + (\frac{m}{t^2}\sigma_f^2(d_2))\sum_{k=1}^{K}\eta_k^2\right)$$

$$\leq \frac{1}{K \eta_K}\left(\frac{\Phi_1 - \Phi^*}{\gamma c_l} + \frac{17}{4K^2}\int\frac{t}{(m+k)^{1/2}} + \frac{4}{3K^2}\int\frac{(m+k)^{1/2}}{t} + (\frac{m}{t^2}\sigma_f^2(d_2))\int\frac{t^2}{m+k}\right)$$

$$\leq \frac{(m+K)^{1/2}}{Kt}\left(\frac{\Phi_1 - \Phi^*}{\gamma c_l} + \frac{17t}{4K^2}(m+K)^{1/2} + \frac{4}{3tK^2}(m+K)^{3/2} + (m\sigma_f^2(d_2))t^2 \ln(m+K)\right)$$

$$\tag{103}$$

According to the Jesen's inequality, we have

$$\frac{1}{K}\sum_{k=1}^{K}\mathbb{E}[\frac{1}{2}\left(\sqrt{2} L_0^2 \|y^*(x_k) - y_k\| + \sqrt{2}\|\nabla f_\delta(x_k, y_k) - w_k - R_k\| + \|R_k\| + \frac{1}{\gamma c_l}\|\tilde{x}_{k+1} - x_k\|\right)]$$

$$\leq \left(\frac{4}{K}\sum_{k=1}^{K}\frac{1}{4}\left(2 L_0^2 \|y^*(x_k) - y_k\|^2 + 2\|\nabla f_\delta(x_k, y_k) - w_k - R_k\|^2 + \|R_k\|^2 + \frac{1}{\gamma^2 c_l^2}\|\tilde{x}_{k+1} - x_k\|^2\right)\right)^{1/2}$$

$$\leq \frac{2(m+K)^{1/4}}{\sqrt{Kt}}\sqrt{\frac{\Phi_1 - \Phi^*}{\gamma c_l} + \frac{17t}{4K^2}(m+K)^{1/2} + \frac{4}{3tK^2}(m+K)^{3/2} + (m\sigma_f^2(d_2))t^2 \ln(m+K)}$$

$$\leq \frac{2m^{1/4}\sqrt{G}}{\sqrt{Kt}} + \frac{2\sqrt{G}}{(Kt)^{1/4}}$$

$$\tag{104}$$

where $G = \dfrac{\Phi_1 - \Phi^*}{\gamma c_l} + \dfrac{17t}{4K^2}(m+K)^{1/2} + \dfrac{4}{3tK^2}(m+K)^{3/2} + (m\sigma_f^2(d_2))t^2 \ln(m+K)$. Finally, we can obtain

$$\frac{1}{K}\sum_{k=1}^{K}\mathbb{E}[\frac{1}{2}\mathcal{M}_k] \leq \frac{2m^{1/4}\sqrt{G}}{\sqrt{Kt}} + \frac{2\sqrt{G}}{(Kt)^{1/4}} + \frac{\delta L}{4\mu_g}C_{gxy}(d_2 + 3)^{3/2}C_{fy}(1 + \frac{1}{\mu_g}(1 - \eta\mu_g)).$$

$$\tag{105}$$

$\square$

