# OpenReview forum: "Double Momentum Method for Lower-Level Constrained Bilevel Optimization"
_ICLR.cc/2024/Conference — ICLR 2024 Conference Withdrawn Submission_

### Official Review · Reviewer_RgCX · 2023-10-19

**Soundness:** 2 fair
**Presentation:** 2 fair
**Contribution:** 3 good
**Rating:** 5
**Confidence:** 3

**Summary:**

This paper considers bi-level optimization problems with constrained lower-level problems (LCBO). A single-loop method is proposed to solve the LCBO, which returns an approximately stationary point with a non-asymptotic convergence rate. The main technique is to use the Gaussian smoothing to approximate the hypergradients. Moreover, momentum methods are applied to update both the upper and lower-level variables. Numerical experiments show the superiority of the proposed method.

**Strengths:**

1. The proposed method is a single-loop algorithm, which is more efficient than the existing methods.
2. The application of the Gaussian smoothing is new to me. This provides new insights for solving the LCBO.

**Weaknesses:**

1. The technical analysis is not sound. It is not clear why $F$ is differentiable (e.g., in Lemma 5) and how Assumption 3 works. Indeed, for a simple example of LCBO, $F$ can be non-differentiable at some points. For example, $F(x)=|x|$ in the following problem is not differentiable at $x=0$
$$
\min_{x,y} -xy \text{ s.t. } y\in\arg\min_{z\in[-1,1]}xz.
$$
However, these kinds of cases are not discussed in the paper. As a consequence, Lemma 5 and (10) are not convincing.

2. Assumption 4 is also strange to me. More discussions are needed for this kind of boundedness assumption.

**Questions:**

1. Why is the convergence rate only related to $d_2$ but not $d_1$?

2. In the experiments, are the lower-level constraints active at the solution returned by the proposed algorithm?

**Details Of Ethics Concerns:**

Non

---

> ### Author Response · Authors · 2023-11-14
> **Respone to reviewer RgCX**
>
> We sincerely appreciate the time you dedicated to reviewing our paper, and we are grateful for the valuable insights you provided. In the sections below, we address each of your questions with careful consideration and thorough responses.
>
> **Q1: It is not clear why is differentiable (e.g., in Lemma 5) and how Assumption 3 works.**
>
> A1: We acknowledge that the assumptions made in our study might have been overly stringent, potentially affecting the applicability of our research. The reason for employing these assumptions was to simplify the problem and facilitate theoretical analysis. However, we recognize that these strong assumptions may pose limitations in specific contexts.  We have modified Assumption 3 in our new version as follows,
> >**Assumption 3.** a) If projection operator has a closed-form solution and **$z^\*=y^\*(x)-\eta \nabla_y g(x,y^\*(x))$ is not on the boundary of the constraint**, then $P_Y(z^*)$ is continuously differentiable in a neighborhood of $z^*$. In addition, in the neighborhood of $z^*$, $P_Y(z^*)$ has Lipschitz continuous gradient with constant $L$. **b) $y^\*(x)$ is continuously differentiable on a neighborhood of $x$**.
>
> In many complicated machine learning problems, the probability that $z^*$ falls exactly on the constraint boundary is very low. $z^*$ primarily reside either within or outside the boundary, i.e., a) the constraint is not active and $\nabla_y g(x,y^*)=0$; b) $y^*$ is on the boundary and $||\nabla_y g(x,y^*)||>0$. In these two cases,  the projection operator could be differentiable in the neighborhood of $z^*$, since we have the closed form of the projection operator and $z^*$ is not on the nonsmooth point.
>
> This assumption is used in [1,2] and [1,2] use this assumption derive the hypergradient (6). Even [1,2] assume $y^*(x)$ and $\mathcal{P}_ {\mathcal{Y}}(z^*)$ are differentiable in a small neighborhood, [1,2] cannot give a convergence analysis. Because they need $||\nabla \mathcal{P}_ {\mathcal{Y}}(z_1^*)-\nabla \mathcal{P}_ {\mathcal{Y}}(z_2^*)||\leq L_ P||z_1^*-z_2^*||$, which is not satisfied. Using the new Assumption 3, we can obtain the gradient of $F$ and ignore the non-differentiable point.
>
> Most importantly, the main purpose and contribution of our paper is to propose a single-loop method with convergence analysis based on [1,2], instead of proposing a new method to calculate the hypergradient. Therefore, even if the hypothesis is relatively strong, we believe that the proposal of this method is still meaningful.
>
> **Q2: Assumption 4 is also strange to me. More discussions are needed for this kind of boundedness assumption.**
>
> A2: For the hypergradient estimation $\nabla f_{\delta}(x,y)$, we have
> >$$
> || \nabla f_ {\delta}(x,y)|| \\
>     \leq || \nabla_x f(x,y)||+ \eta ||\nabla_ {xy}^ 2 g(x,y) || \cdot||\nabla\mathcal{P}_ {\mathcal{Y}\delta}(z)^ {\top}||\cdot ||\sum_ {i=0}^ {Q-1}\left((I_ {d_2}-\eta\nabla_ {yy}^2 g(x,y))\nabla\mathcal{P}_ {\mathcal{Y}\delta}(z)^{\top}\right)^i||\cdot||\nabla_yf(x,y)||\\
>     =C_{fx}+\eta C_{gxy}C_{fy}\sum_ {i=0}^ {Q-1}||I_ {d_2}-\eta\nabla_ {yy}^ 2g(x,y)||^i\\
>     \leq C_ {fx}+\dfrac{C_ {gxy}C_ {fy}}{\mu_ g}
> $$
>
> Then, using the variance in Lemma 4, we can bound the norm of stochastic estimation, i.e.,  $||\bar{\nabla} f_{\delta}(x,y;\bar{\xi})||\leq C$. Therefore, we can easily obtain $||w||$ is bounded, since $w_{k+1}=(1-\alpha)w_{k}+\alpha\bar{\nabla} f_{\delta}(x_{k+1},y_{k+1};\bar{\xi}_{k+1})$ and $0<\alpha<1$. Similarly, we can bound $v$. Even if the norm of the gradient estimation is not bounded, we can also use the clipping method to make the norm of $v$ and $u$ bounded and finally make Assumption 4 hold.

---

> ### Author Response · Authors · 2023-11-17
> **Respone to reviewer RgCX part 2**
>
> **Q3: Why is the convergence rate only related to $d_2$ but not $d_1$?**
>
> A3: Since we use the zeroth order method to approximate the Jacobian matrix of the projection operator, the Lipschitz constant and variance of the approximation of the hypergradient is related to $d_2$. In addition, from our analysis, we can find that the convergence rate highly depends on these constants which make the convergence rate related to $d_2$. However, these constant does not affect these constants in the approximation of the hypergradient and therefore the convergence rate is not related to $d_1$.
>
> **Q4: Are the lower-level constraints active at the solution returned by the proposed algorithm?**
>
> A4: Since the dimension of the lower-level variable is relatively large and the constraint parameter $r=1$ is relatively small,  using gradient update in $y$ will easily make $y$ get out of the constraint and lead to the projection on the boundary of the constraints. This makes the lower-level constraints active at the solution return by our method. We can easily check the constraint at the end of our algorithm.
>
> [1] Blondel M, Berthet Q, Cuturi M, et al. Efficient and modular implicit differentiation[J]. Advances in neural information processing systems, 2022, 35: 5230-5242.
> [2] Bertrand Q, Klopfenstein Q, Massias M, et al. Implicit differentiation for fast hyperparameter selection in non-smooth convex learning[J]. The Journal of Machine Learning Research, 2022, 23(1): 6680-6722.
> [3] Guo Z, Xu Y, Yin W, et al. On stochastic moving-average estimators for non-convex optimization. ArXiv e-prints[J]. arXiv preprint arXiv:2104.14840, 2021.
> [4] Shi W, Gao H, Gu B. Gradient-Free Method for Heavily Constrained Nonconvex Optimization[C]//International Conference on Machine Learning. PMLR, 2022: 19935-19955.
> [5] Guo Z, Xu Y, Yin W, et al. A novel convergence analysis for algorithms of the adam family and beyond[J]. arXiv preprint arXiv:2104.14840, 2021.

---

> > ### Author Response · Authors · 2023-11-22
> > **looking forward to post-rebuttal feedback!**
> >
> > Dear Reviewer RgCX
> >
> > Thank you for reviewing our paper. We have carefully answered your concerns about the theorem with dimension d and assumptions. Please let us know if our answers accurately address your concerns. If our response resolves your concerns, we kindly ask you to consider raising the rating of our work. Thank you very much for your time and efforts! We would like to discuss any additional questions you may have.
> >
> > Best,
> > Authors

---

> > > ### Comment · Reviewer_RgCX · 2023-11-22
> > >
> > > Thanks for your reply. Most of my concerns are addressed. However, the assumptions are still not satisfactory to me. The new assumptions are not easy to verify, either. For example, how can we ensure that $y^*(x)$ is differentiable? I think this is one of the core problems in Bi-level optimization and assuming the differentiability of $y^*(x)$ is improper. As a consequence, the developed theoretical results seem not to be solid. Hence, I maintain my score.
> > >
> > > It is suggested that the authors build their theory on standard conditions in existing literature, e.g., the PL condition. This may improve the theoretical contributions of this paper.

---

### Official Review · Reviewer_voeq · 2023-10-30

**Soundness:** 2 fair
**Presentation:** 3 good
**Contribution:** 3 good
**Rating:** 5
**Confidence:** 4

**Summary:**

To address lower-level constrained bilevel optimization problem, the authors leverage the Gaussian smoothing to approximate the hypergradent. Furthermore, the author proposes a single-loop single-timescale algorithm and theoretically prove its convergence rates. Two experimental settings have been tested to demonstrate the superiority of proposed algorithm.

**Strengths:**

The experimental results are great for proposed algorithm.

**Weaknesses:**

1. The proposed algorithm DMLCBO is based on double momentum technique. In previous works, e.g., SUSTAIN[1] and MRBO[2], double momentum technique improves the convergence rate to $\mathcal{\widetilde O}(\epsilon^{-3})$ while proposed algorithm only achieves the $\mathcal{\widetilde O}(\epsilon^{-4})$. The authors are encouraged to discuss the reason why DMLCBO does not achieve it and the theoretical technique difference between DMLCBO and above mentioned works.

2. In the experimental part, the author only shows the results of DMLCBO in early time, it will be more informative to provide results in the later steps.

3. In Table 3, DMLCBO exhibits higher variance compared with other baselines in MNIST datasets, the authors are encouraged to discuss more experimental details about it and explain the behind reason.


[1] A Near-Optimal Algorithm for Stochastic Bilevel Optimization via Double-Momentum
[2] Provably Faster Algorithms for Bilevel Optimization

**Questions:**

Check the weakness part.

---

> ### Author Response · Authors · 2023-11-14
> **Respone to reviewer voeq**
>
> We sincerely appreciate the time you dedicated to reviewing our paper, and we are grateful for the valuable insights you provided. In the sections below, we address each of your questions with careful consideration and thorough responses.
>
> **Q1: The authors are encouraged to discuss the reason why DMLCBO does not achieve $\tilde{O}(\epsilon^{-3})$ and the theoretical technique difference between DMLCBO and SUSTAIN[1] and MRBO[2].**
>
> A1: Your comments are greatly appreciated. SUSTAIN[1] and MRBO[2] using the Variance Reduction methods, i.e., STORM and SPIDER, both on the gradient estimation of $x$ and $y$.  These methods need stochastic gradient to be Lipschitz continuous. However, this condition is not satisfied in our method. Therefore, we can only use the traditional moving average method which leads to a different convergence rate. Similar results can be found (i.e., $\tilde{\mathcal{O}}(\epsilon^{-4})$) in [3,4] if the stochastic gradient is Lipschitz continuous.
>
> **Q2: The author only shows the results of DMLCBO in early time, it will be more informative to provide results in the later steps.**
>
> A2: We highly value your feedback and comments. Please be aware that we've already included results in subsequent sections within our appendix. We extended the iteration count of our method to 100,000 to illustrate its convergence performance. Thank you for your consideration!
>
> **Q3: DMLCBO exhibits higher variance compared with other baselines in MNIST datasets.**
> A3: Your comments are greatly appreciated. In our method, we use the zeroth order method which may have a large variance. This makes the poison data points found by our methods have high variance in different turns. And MNIST is a simple dataset and easily to be attacked which will increase the variance in the poison data points. Finally, these variances will highly affect the retrained model and lead to a large variance in the test accuracy.
>
> [1] A Near-Optimal Algorithm for Stochastic Bilevel Optimization via Double-Momentum
>
> [2] Provably Faster Algorithms for Bilevel Optimization
>
> [3] Dagréou M, Ablin P, Vaiter S, et al. A framework for bilevel optimization that enables stochastic and global variance reduction algorithms[J].
>
> [4] Chen X, Xiao T, Balasubramanian K. Optimal Algorithms for Stochastic Bilevel Optimization under Relaxed Smoothness Conditions[J].

---

> > ### Author Response · Authors · 2023-11-22
> > **looking forward to post-rebuttal feedback!**
> >
> > Dear Reviewer voeq
> >
> > Thank you for reviewing our paper. We have carefully answered your concerns about the experiments and theorem. Please let us know if our answers accurately address your concerns. If our response resolves your concerns, we kindly ask you to consider raising the rating of our work. Thank you very much for your time and efforts! We would like to discuss any additional questions you may have.
> >
> > Best,
> > Authors

---

### Official Review · Reviewer_VJdq · 2023-11-01

**Soundness:** 2 fair
**Presentation:** 3 good
**Contribution:** 2 fair
**Rating:** 5
**Confidence:** 3

**Summary:**

This paper studies a bilevel optimization problem in which the lower-level problem has a convex set constraint which is independent of the upper-level variable. Using Gaussian smoothing to approximate the gradient of the projection operator, the authors propose an approximation to the hypergradient and a single-loop algorithm. Theoretical analysis and numerical experiments are provided.

**Strengths:**

The proposed algorithm is a single-loop single-timescale approach.

**Weaknesses:**

1. Assumption 3 is restrictive to satisfy. Furthermore, even the problems examined in the numerical experiments fail to meet this assumption.

2. In order to achieve a stationary point with $\|| \nabla F (x) \|| \le \epsilon$, as outlined in Remark 2, the proposed algorithm necessitates a choice of the smooth parameter on the order of $O(\epsilon d_2^{-3/2})$. Consequently, the algorithm would require a minimum of approximately $\tilde{O}(d_2^8/\epsilon^8)$ iterations. It appears, however, that the authors aim to obscure this fact within their paper and retain the smooth parameter in their complexity result.

3. The problems explored in the numerical experiments may not necessarily adhere to the strongly convex assumption for the lower-level problem stipulated in Assumption 2 (3). Moreover, the selection of values for the parameters $Q$ and $\eta$ does not align with the theoretical requirements specified in Theorem 1.

**Questions:**

see above

---

> ### Author Response · Authors · 2023-11-14
> **Respone to reviewer VJdq**
>
> We sincerely appreciate the time and effort you dedicated to reviewing our paper, and we are thankful for your constructive comments. In the sections below, we address each of the questions you raised.
>
> **Q1: Assumption 3 is restrictive to satisfy.**
>
> A1: We acknowledge that the assumptions made in our study might have been overly stringent, potentially affecting the applicability of our research. The reason for employing these assumptions was to simplify the problem and facilitate theoretical analysis. However, we recognize that these strong assumptions may pose limitations in specific contexts.  We have modified Assumption 3 in our new version as follows,
> >**Assumption 3.** a) If projection operator has a closed-form solution and **$z^\*=y^\*(x)-\eta \nabla_y g(x,y^\*(x))$ is not on the boundary of the constraint**, then $P_Y(z^*)$ is continuously differentiable in a neighborhood of $z^*$. In addition, in the neighborhood of $z^*$, $P_Y(z^*)$ has Lipschitz continuous gradient with constant $L$. **b) $y^\*(x)$ is continuously differentiable on a neighborhood of $x$**.
>
> In many complicated machine learning problems, the probability that $z^*$ falls exactly on the constraint boundary is very low. $z^*$ primarily reside either within or outside the boundary, i.e., a) the constraint is not active and $\nabla_y g(x,y^*)=0$; b) $y^*$ is on the boundary and $||\nabla_y g(x,y^*)||>0$. In these two cases,  the projection operator could be differentiable in the neighborhood of $z^*$, since we have the closed form of the projection operator and $z^*$ is not on the nonsmooth point.
>
>
> This assumption is used in [1,2] and [1,2] use this assumption derive the hypergradient (6). However, these methods do not have the convergence analysis. The main purpose and contribution of our paper is to design a single-loop method with convergence analysis based on [1,2], instead of proposing a new method to calculate the hypergradient. Therefore, even if the hypothesis is relatively strong, we believe that the proposal of our method is still meaningful.
>
> **Q2:the algorithm would require a minimum of approximately $\mathcal{O}(d_2^8\epsilon^{-8})$ iterations.**
>
> A2:  Thank you very much for pointing out our mistake. We have modified the proof in the new version, hope you can check it again.
>
> In our new proof of Theorem 1, we use the following new function,
> >$$
> \Phi_{k+1}\\
>         =\mathbb{E}[F_{\delta}(x_{k+1})+\dfrac{10L_0^a c_l}{\tau\mu_g c_u}||y_{k+1}-y^*(x_{k+1})||^2 + c_l(\|w_{k+1}-\bar{\nabla} f_{\delta}(x_{k+1},y_{k+1})-R_{k+1}\|^2\\
>         +\|\nabla_y g(x_{k+1},y_{k+1})-v_{k+1}\|^2)]
> $$
>
> where $0\leq a$ is a constant. We can obtain the following result.
> >$$
> \dfrac{1}{K}\sum_{k=1}^ K\mathbb{E}[\dfrac{1}{2}\mathcal{M}_ k]
>         \leq\dfrac{2m^{1/4}\sqrt{G}}{\sqrt{Kt}}+\dfrac{2\sqrt{G}}{(Kt)^{1/4}}+\dfrac{\delta L}{4\mu_g} C_ {gxy}(d_2+3)^{3/2}C_ {fy}(1+\dfrac{1}{\mu_g}(1-\eta\mu_g)).
> $$
>
> where $G=\dfrac{\Phi_1-\Phi^*}{\gamma c_l}+\dfrac{17t}{4K^2}(m+K)^{1/2}+\dfrac{4}{3tK^2}(m+K)^{3/2}+(m\sigma_{f}^2(d_2))t^2\ln (m+K)$.
> Then, setting $a=0$, we have $\sqrt{G}=\tilde{\mathcal{O}}(\sqrt{d_ 2})$.  Let $\delta=\mathcal{O}(\epsilon d_2^ {-3/2})$, $r$ randomly sampled from $\{0,1,\cdots,K\}$, we have $\mathbb{E}[\dfrac{1}{2}\mathcal{M}_ r]=\dfrac{1}{K}\sum_ {k=1}^ K\mathbb{E}[\dfrac{1}{2}\mathcal{M}_k]=\tilde{\mathcal{O}}(\dfrac{\sqrt{d_2}}{ K^{1/4}})\leq \epsilon$. Therefore, we have $K=\tilde{\mathcal{O}}(\dfrac{d_2^2}{ \epsilon^{4}})$. This is the first method with convergence analysis for LCBO with general constraints, which directly solves the BO problem. Comparatively, in [6], for the single-level nonsmooth problem, the authors derived an iteration number of $\mathcal{O}(d^{3/2}\delta^{-1}\epsilon^{-4})$. This outcome demonstrates that, even within the LCBO context, our method achieves results akin to those in the **single-level** problem setting."

---

> > ### Author Response · Authors · 2023-11-19
> > **Respone to reviewer VJdq part 2**
> >
> > **Q3: The problems explored in the numerical experiments may not necessarily adhere to the strongly convex assumption for the lower-level problem stipulated in Assumption 2 (3). Moreover, the selection of values for the parameters $\eta$ and $Q$ does not align with the theoretical requirements specified in Theorem 1.**
> >
> >
> > A3: According to Theorem 1, the optimal $Q$ could be larger than that we used in our experiments. However, as we show in our appendix, increasing $Q$ does not significantly improve accuracy, but will increase the time of our method. Therefore, It is reasonable to choose a smaller $Q$ and the errors can be tolerated. We also study the effect of $\eta$ in our appendix and we can find the errors using different $\eta$ can be tolerated. The experiments used in our paper are very common in bilevel optimization optimizations [3,4,5]. Even [3,4,5] assume that the lower-level problem is strongly convex, the lower-level problems considered in their experiments are still non-convex. In addition, evaluating all the methods on the nonconvex lower-level problem can more generally illustrate the effectiveness of the algorithm even if it lacks convergence analysis for the non-convex lower-level problem.
> >
> >
> > [1] Blondel M, Berthet Q, Cuturi M, et al. Efficient and modular implicit differentiation[J]. Advances in neural information processing systems, 2022, 35: 5230-5242.
> > [2] Bertrand Q, Klopfenstein Q, Massias M, et al. Implicit differentiation for fast hyperparameter selection in non-smooth convex learning[J]. The Journal of Machine Learning Research, 2022, 23(1): 6680-6722.
> > [3] Khanduri P, Zeng S, Hong M, et al. A near-optimal algorithm for stochastic bilevel optimization via double-momentum[J]. Advances in neural information processing systems, 2021, 34: 30271-30283.
> > [4] Chen T, Sun Y, Xiao Q, et al. A single-timescale method for stochastic bilevel optimization[C]//International Conference on Artificial Intelligence and Statistics. PMLR, 2022: 2466-2488.
> > [5] Hong M, Wai H T, Wang Z, et al. A two-timescale stochastic algorithm framework for bilevel optimization: Complexity analysis and application to actor-critic[J]. SIAM Journal on Optimization, 2023, 33(1): 147-180.
> > [6] Lin T, Zheng Z, Jordan M. Gradient-free methods for deterministic and stochastic nonsmooth nonconvex optimization[J]. Advances in Neural Information Processing Systems, 2022, 35: 26160-26175.

---

> > > ### Author Response · Authors · 2023-11-22
> > > **looking forward to post-rebuttal feedback!**
> > >
> > > Dear Reviewer VJdq
> > >
> > > Thank you for reviewing our paper. We have carefully answered your concerns on the assumptions, theorem, and experimental settings. Please let us know if our answers accurately address your concerns. If our response resolves your concerns, we kindly ask you to consider raising the rating of our work. Thank you very much for your time and efforts! We would like to discuss any additional questions you may have.
> > >
> > > Best,
> > > Authors

---

### Official Review · Reviewer_AArG · 2023-11-06

**Soundness:** 2 fair
**Presentation:** 3 good
**Contribution:** 2 fair
**Rating:** 5
**Confidence:** 3

**Summary:**

The authors introduce a novel hypergradient approximation method for lower-level constrained bilevel optimization problems with non-asymptotic convergence analysis, utilizing Gaussian smoothing. This method incorporates double-momentum and adaptive step size techniques. The experimental results, in the context of data hyper-cleaning and training data poisoning attacks, showcase the efficiency and effectiveness of the proposed approach.

**Strengths:**

S1. The work is well motivated. Finding a simple yet effective method for lower-level constrained bilevel optimization problems is both interesting and important.

S2. The paper is well written and easy to follow. The algorithm design is new and non-asymptotic convergence is provided.

S3. The authors conduct numerous experiments to showcase the efficiency and effectiveness of the proposed approach. Additionally, the paper includes several ablation studies in the Appendix.

**Weaknesses:**

W1. By Remark 2 on page 7, $\tilde{\mathcal{O}}(\frac{\sqrt{d_2}}{\delta K^{1/4}})\leq \epsilon$ implies that $K=\tilde{\mathcal{O}}(\frac{d_2^2}{\delta^4 \epsilon^4})$, NOT $\tilde{\mathcal{O}}(\frac{\sqrt{d_2}}{\delta \epsilon^4})$. Additionally, since $\delta=\mathcal{O}(\epsilon d_2^{-3/2})$ by (11), the iteration number $K=\tilde{\mathcal{O}}(\frac{d_2^8}{\epsilon^8})$.

W2. The authors should consider comparing their method with closely related papers addressing lower-level constrained bilevel optimization problems, including:

[1] Han Shen, Tianyi Chen. “On Penalty-based Bilevel Gradient Descent Method.” ICML 2023.

W3. Since there is an additional loop to approximate the matrix inverse, it can be noted that the proposed algorithm DMLCBO is not fully single-loop.

**Questions:**

Q1. Could you provide some representative class of problems that satisfy Assumption 3? Consider the simple example: $g(x,y)=(y-x)^2/2$, $\mathcal{Y}=[-1,1]$ and $\mathcal{X}=[-3,3]$. The projection operator $P_Y$ has a closed-form solution, but $\mathcal{P}_{\mathcal{Y}}(z^*)$ is not continuously differentiable in a neighborhood of $z^*$ when $x=1$ or $x=-1$.

Q2. Is Assumption 3 satisfied for all small values of $\eta$?

Q3. What measures can be taken to verify that Assumption 4 is satisfied, or are there specific checkable sufficient conditions to ensure its validity?

Minor Comments:

(1)On page 3, in Equation (4): The minus sign in the expression of $\nabla y^*(x)$ was omitted.

(2)On page 4, in Remark 1: What is $F_{\delta}(x)$?

(3)On page 5, in Lemma 2: “$\| A \| \leq 1$” should be “$\| A \| < 1$”. Make similar changes after Lemma 2.

(4)On page 5, in Equation (9): By the definition $c(Q)$, the term $u^Q$ in $\bar{\xi}$ is not used.

(5)On page 6, in Algorithm 1: swap the positions of $v_1$ and $w_1$. Should “$g(x_1,y_1)$" be replaced with “$\nabla_y g(x_1,y_1)$"? Make similar changes in Section 3.3.

(6)On page 6, in line 3 from below: Should “$\nabla F_{\delta}(x_k)$” be replaced with “$\nabla F (x_k)$”?

(7)On page 7, Lemma 5: The absolute value symbol for $\mathcal{G}$ in Equation (10) was omitted. Should “$\nabla f(x_k, y_k)$” be replaced with “$\nabla_x f(x_k, y_k)$”? Additionally, what is $\tilde{x}_{k+1}$?

(8)On page 8, in line 3 from below: correct the sum within the max part.

(9)On page 9, Do the bilevel optimization problems related to training data poisoning attacks satisfy the smoothness and convexity assumptions?Note that “a network with two convolution layers and two fully-connected-layer layers for MNIST and a network with three convolution layers and three fully-connected-layer layers for Cifar10, where the Relu function is used in each layer.”

---

> ### Author Response · Authors · 2023-11-14
> **Respone to Reviewer AArG**
>
> We are grateful for your time dedicated to reviewing our paper, as well as your constructive comments. Below we address each question.
>
> **Q1: By Remark 2 on page 7, the iteration number $K=\mathcal{O}(\dfrac{d_2^8}{\epsilon^8})$**.
>
> A1: Thank you very much for pointing out our mistake. We have modified the proof in the new version, hope you can check it again.
>
> In our new proof of Theorem 1, we use the following new function,
> >$$
> \Phi_{k+1}\\
>         =\mathbb{E}[F_{\delta}(x_{k+1})+\dfrac{10L_0^a c_l}{\tau\mu_g c_u}||y_{k+1}-y^\*(x_{k+1})||^2 + c_l(\|w_{k+1}-\bar{\nabla} f_{\delta}(x_{k+1},y_{k+1})-R_{k+1}\|^2\\
>         +\|\nabla_y g(x_{k+1},y_{k+1})-v_{k+1}\|^2)]
> $$
>
> where $0\leq a$ is a constant. We can obtain the following result.
> >$$
> \dfrac{1}{K}\sum_{k=1}^ K\mathbb{E}[\dfrac{1}{2}\mathcal{M}_ k]
>         \leq\dfrac{2m^{1/4}\sqrt{G}}{\sqrt{Kt}}+\dfrac{2\sqrt{G}}{(Kt)^{1/4}}+\dfrac{\delta L}{4\mu_g} C_ {gxy}(d_2+3)^{3/2}C_ {fy}(1+\dfrac{1}{\mu_g}(1-\eta\mu_g)).
> $$
>
> where $G=\dfrac{\Phi_1-\Phi^*}{\gamma c_l}+\dfrac{17t}{4K^2}(m+K)^{1/2}+\dfrac{4}{3tK^2}(m+K)^{3/2}+(m\sigma_{f}^2(d_2))t^2\ln (m+K)$.
> Then, setting $a=0$, we have $\sqrt{G}=\tilde{\mathcal{O}}(\sqrt{d_ 2})$.  Let $\delta=\mathcal{O}(\epsilon d_2^ {-3/2})$, $r$ randomly sampled from $\{0,1,\cdots,K\}$, we have $\mathbb{E}[\dfrac{1}{2}\mathcal{M}_ r]=\dfrac{1}{K}\sum_ {k=1}^ K\mathbb{E}[\dfrac{1}{2}\mathcal{M}_k]=\tilde{\mathcal{O}}(\dfrac{\sqrt{d_2}}{ K^{1/4}})\leq \epsilon$. Therefore, we have $K=\tilde{\mathcal{O}}(\dfrac{d_2^2}{ \epsilon^{4}})$. This is the first method with convergence analysis for LCBO with general constraints, which directly solves the BO problem. Comparatively, in [13], for the single-level nonsmooth problem, the authors derived an iteration number of $\mathcal{O}(d^{3/2}\delta^{-1}\epsilon^{-4})$. This outcome demonstrates that, even within the LCBO context, our method achieves results akin to those in the **single-level** problem setting."
>
>
> **Q2: Could you provide some representative class of problems that satisfy Assumption 3? Is Assumption 3 satisfied for all small values of $\eta$?**
>
> A2: We acknowledge that the assumptions made in our study might have been overly stringent, potentially affecting the applicability of our research. The reason for employing these assumptions was to simplify the problem and facilitate theoretical analysis. However, we recognize that these strong assumptions may pose limitations in specific contexts.  We have modified Assumption 3 in our new version as follows,
> >**Assumption 3.** a) If projection operator has a closed-form solution and **$z^\*=y^\*(x)-\eta \nabla_y g(x,y^\*(x))$ is not on the boundary of the constraint**, then $P_Y(z^*)$ is continuously differentiable in a neighborhood of $z^*$. In addition, in the neighborhood of $z^*$, $P_Y(z^*)$ has Lipschitz continuous gradient with constant $L$. **b) $y^\*(x)$ is continuously differentiable on a neighborhood of $x$**.
>
> In many complicated machine learning problems, the probability that $z^*$ falls exactly on the constraint boundary is very low. $z^*$ primarily reside either within or outside the boundary, i.e., a) the constraint is not active and $\nabla_y g(x,y^*)=0$; b) $y^*$ is on the boundary and $||\nabla_y g(x,y^*)||>0$. In these two cases,  the projection operator could be differentiable in the neighborhood of $z^*$, since we have the closed form of the projection operator and $z^*$ is not on the nonsmooth point.
>
>
>
> In addition, this assumption has been used in [4,5] to derive the hypergradient (6). The main contribution of this paper is not to use Assumption 3 to derive the hypergradient. The main purpose and contribution of our paper is to design a single-loop method with convergence analysis based on [4,5]. Therefore, even if the hypothesis is relatively strong, we believe that the proposal of our method is still meaningful. Specifically, we propose a single loop method for the LCBO and our method achieves convergence results akin to that in the **single-level** problem setting.

---

> ### Author Response · Authors · 2023-11-14
> **Respone to Reviewer AArG part2**
>
> **Q3: What measures can be taken to verify that Assumption 4 is satisfied, or are there specific checkable sufficient conditions to ensure its validity?**
>
> A3:  For the hypergradient estimation $\nabla f_{\delta}(x,y)$, we have
>
> >$$
> || \nabla f_ {\delta}(x,y)|| \\
>     \leq || \nabla_x f(x,y)||+ \eta ||\nabla_ {xy}^ 2 g(x,y) || \cdot||\nabla\mathcal{P}_ {\mathcal{Y}\delta}(z)^ {\top}||\cdot ||\sum_ {i=0}^ {Q-1}\left((I_ {d_2}-\eta\nabla_ {yy}^2 g(x,y))\nabla\mathcal{P}_ {\mathcal{Y}\delta}(z)^{\top}\right)^i||\cdot||\nabla_yf(x,y)||\\
>     =C_{fx}+\eta C_{gxy}C_{fy}\sum_ {i=0}^ {Q-1}||I_ {d_2}-\eta\nabla_ {yy}^ 2g(x,y)||^i\\
>     \leq C_ {fx}+\dfrac{C_ {gxy}C_ {fy}}{\mu_ g}
> $$
>
> Then, using the variance in Lemma 4, we can bound the norm of stochastic estimation, i.e.,  $||\bar{\nabla} f_{\delta}(x,y;\bar{\xi})||\leq C$. Therefore, we can easily obtain $||w||$ is bounded, since $w_{k+1}=(1-\alpha)w_{k}+\alpha\bar{\nabla} f_{\delta}(x_{k+1},y_{k+1};\bar{\xi}_{k+1})$ and $0<\alpha<1$. Similarly, we can bound $v$. Even if the norm of the gradient estimation is not bounded, we can also use the clipping method to make the norm of $v$ and $u$ bounded and finally make Assumption 4 hold.
>
> **Q4: Since there is an additional loop to approximate the matrix inverse, it can be noted that the proposed algorithm DMLCBO is not fully single-loop.**
>
> A4: In bilevel optimization, the algorithm that alternately updates two variables is referred to as a single-loop algorithm even if it needs to approximate the matrix inverse, such as [1,2,3].
>
> **Q5: Do the bilevel optimization problems related to training data poisoning attacks satisfy the smoothness and convexity assumptions?**
>
> A5:
> Thank you for bringing to our attention the issue in our paper. We acknowledge that our assumption regarding convexity and smoothness does not align with the lower-level problem in the training data poisoning attacks, particularly due to the use of DNNs with ReLU functions. Consequently, the theoretical analysis provided may not be directly applicable in this scenario. Nevertheless, it's essential to note that while our theoretical framework might not directly apply, it doesn't invalidate the capability of our algorithm to address this specific problem.
>
> For other typos, we have corrected in our new version, and $\tilde{x}_ {k+1}=P_X(x_k-\dfrac{\gamma}{\sqrt{||w_{k}||}+G_0}w_k)$.
>
>
> [1] Khanduri P, Zeng S, Hong M, et al. A near-optimal algorithm for stochastic bilevel optimization via double-momentum[J]. Advances in neural information processing systems, 2021, 34: 30271-30283.
>
> [2] Chen T, Sun Y, Xiao Q, et al. A single-timescale method for stochastic bilevel optimization[C]//International Conference on Artificial Intelligence and Statistics. PMLR, 2022: 2466-2488.
>
> [3] Hong M, Wai H T, Wang Z, et al. A two-timescale stochastic algorithm framework for bilevel optimization: Complexity analysis and application to actor-critic[J]. SIAM Journal on Optimization, 2023, 33(1): 147-180.
>
> [4] Blondel M, Berthet Q, Cuturi M, et al. Efficient and modular implicit differentiation[J]. Advances in neural information processing systems, 2022, 35: 5230-5242.
>
> [5] Bertrand Q, Klopfenstein Q, Massias M, et al. Implicit differentiation for fast hyperparameter selection in non-smooth convex learning[J]. The Journal of Machine Learning Research, 2022, 23(1): 6680-6722.
>
> [6] Guo Z, Xu Y, Yin W, et al. On stochastic moving-average estimators for non-convex optimization. ArXiv e-prints[J]. arXiv preprint arXiv:2104.14840, 2021.
>
> [7] Shi W, Gao H, Gu B. Gradient-Free Method for Heavily Constrained Nonconvex Optimization[C]//International Conference on Machine Learning. PMLR, 2022: 19935-19955.
>
> [8] Guo Z, Xu Y, Yin W, et al. A novel convergence analysis for algorithms of the Adam family and beyond[J]. arXiv preprint arXiv:2104.14840, 2021.
>
> [9] Mehra A, Hamm J. Penalty method for inversion-free deep bilevel optimization[C]//Asian Conference on Machine Learning. PMLR, 2021: 347-362.
>
> [10] Shi W, Gu B. Improved Penalty Method via Doubly Stochastic Gradients for Bilevel Hyperparameter Optimization[C]//Proceedings of the AAAI Conference on Artificial Intelligence. 2021, 35(11): 9621-9629.
>
> [11] Yuanzhi Li and Yang Yuan. Convergence analysis of two-layer neural networks with relu activation.
> In Advances in Neural Information Processing Systems, pp. 597–607, 2017
>
> [12] Charles Z, Papailiopoulos D. Stability and generalization of learning algorithms that converge to global optima[C]//International conference on machine learning. PMLR, 2018: 745-754.
>
> [13] Lin T, Zheng Z, Jordan M. Gradient-free methods for deterministic and stochastic nonsmooth nonconvex optimization[J]. Advances in Neural Information Processing Systems, 2022, 35: 26160-26175.

---

> > ### Author Response · Authors · 2023-11-22
> > **looking forward to post-rebuttal feedback!**
> >
> > Dear Reviewer AArG
> >
> > Thank you for reviewing our paper. We have carefully answered your concerns on the assumptions and theorem. Please let us know if our answers accurately address your concerns. If our response resolves your concerns, we kindly ask you to consider raising the rating of our work. Thank you very much for your time and efforts! We would like to discuss any additional questions you may have.
> >
> > Best,
> > Authors

---

### Meta-Review · Area_Chair_vG8m · 2023-12-15

**Metareview:**

This paper proposes a single-loop, single-timescale method for solving lower-level constrained bilevel optimization problems and provides a theoretical analysis of its complexity. One concern that is shared by several reviewers is the practical relevance of the assumptions. Another concern is the technical correctness of the complexity result. Although the authors provide a response to this concern, the subsequent correction will require another round of review. Based on the above, I regrettably have to reject the paper.

**Justification For Why Not Higher Score:**

Non-standard and restrictive assumption is used to develop the theoretical results. Moreover, the technical correctness of the complexity result is not completely resolved.

**Justification For Why Not Lower Score:**

N/A

---

### Decision · Program_Chairs · 2024-01-16

Reject